# Export Decision and Credit Constraints under Institution Obstacles

**Trang Hoai Phan** [1,*] **, Rainer Stachuletz** [2] **and Hai Thi Hong Nguyen** [3]

1   International Economics, Technical University of Darmstadt, Hochschulstr. 1, 64289 Darmstadt, Germany
2   Department of Economics 1, Berlin School of Economics and Law, Badensche Str. 52, 10825 Berlin, Germany; rainer.stachuletz@hwr-berlin.de
3   Faculty of International Business, Banking Academy of Vietnam, Chua Boc Str. 12, Dong Da Dist., 100000 Hanoi, Vietnam; hainth@hvnh.edu.vn
*   Correspondence: trang@vwl.tu-darmstadt.de; Tel.: +49-6151-16-22872

**Abstract:** The growing demand for goods and technology increases capital requirements, especially in exporting enterprises. However, many firms have difficulty accessing external capital due to institutional obstacles. This study analyzes two main issues: the influence of institutional obstacles on credit constraints and the relationship between credit constraints and export decisions, adopting firm-level data from 131 countries. The study's remarkable contribution is to cluster the data into four country groups based on their national income. The typical specification of each group can lead to more precise results, thereby highlighting the role of institutions. More advanced, this study complements the literature's gap in the relationship between credit constraints and exports by controlling for institutions as interactive variables in the model. This work upgrades assessments to be more accurate, thereby providing more valuable information to policymakers. In addition, credit constraints are measured by both quantitative and qualitative methods. The essential role of firm size is emphasized in further analysis. This study approaches the Probit method. Furthermore, an instrumental variable is used to solve the endogeneity problem. The results found that a weak institution prevents access to finance, especially in middle-income countries. In addition, firms' access to capital negatively affects exports in all regions. The finding in the group of rich countries is most pronounced.

**Keywords:** credit constraints; export; institutions; tax rate; political instability; corruption; business licensing and permit; World Bank data 2020; IV-Probit regression

## 1. Introduction

External credit plays an essential role in a firm's operations. This finance allows the firm to use internal financial resources for other purposes, such as cash payments to suppliers or responding to liquidity shocks [1]. In addition, thanks to external capital, firms might carry out projects that they cannot afford on their own, such as production expansion [2], innovation [3], export [4], and investment [5]. As a result, credit contributes to ensuring firms' operation continuity; improving the efficiency of capital utilization; contributing to the formation of the optimal capital structure and the concentration of production capital, improving competitiveness. Moreover, a healthy firm growth might promote the development of the economy, contribute to the renewal of monetary policy, and enhance the operation of credit organizations [6,7].

Despite its role, access to credit is one of the main obstacles to a firm's performance and growth [8], especially in *SMEs* and developing countries. Almost 40% of *SMEs* in developing countries struggled with access to finance, mainly *SMEs* in East Asia and the Asia Pacific. In addition, about half of *SMEs* either do not have access to formal credit from local banks or face unfavorable lending conditions [9]. Numerous studies suggested that the credit constraints problem becomes more serious during international trade opening.

Export sunk costs, logistics costs, and import and export policy regulations are very costly [10]. For example, trade policy barriers such as tariffs and regulations account for at least 14% of the trade costs of Vietnamese exporters (https://wtocenter.vn/an-pham/17975-the-white-book-on-vietnamese-enterprise-2021, accessed on 5 April 2022). Only firms with financial capacity have the opportunity to approach foreign customers [11,12]. Therefore, entering foreign markets creates financial pressure for domestic firms. The linkage between credit constraints and exports is heterogeneous, which is caused by many factors. Jappelli [13] demonstrated that firms' idiosyncrasies influence their capital structure and their ability to access external funding. This ability depends on a firm's inherent factors (such as age, industry, geographical location, size of the firm), financial features (such as the ability to repay debts, the possibility of project success, prestige, collateral), and the owner's characteristics (such as the owner's capacity, network). For example, the firm's plan is unclear, and high risk can negatively affect its credit rating. At the same time, lenders' decisions are based on the borrower's credit rating, such as repayment ability, bad debt history, capital structure, and business risk. As a result, these specifications play a critical role in a firm's ability to access funding [14]. Moreover, Beck [15] emphasized the importance of firm size in accessing external capital. He believed that small firms are struggling more to receive a bank loan than large firms. Thus, financial constraints have more severe consequences for small firms. Moreover, Diagne [16] argued that the credit supply is not infinite. In other words, lenders cannot cover all credit needs. Their ability to provide credit is constrained by external factors such as the structure of the finance sector [17], competition in the banking sector, lack of an effective contract enforcement mechanism, and asymmetric information between lenders and borrowers [18]. These factors play an important role because they determine the price of financial products and affect the willingness to lend [19]. In many cases, the competitiveness of the banking system is dependent on the country's regulation and conditions [20]. Paravisini [21] argued that local Argentine banks do not have a large internal capital market to attract funding in the context that 50% of the banking system's total assets belong to international banks. Thus, domestic banks are more likely to be financially constrained than foreign banks. This result affects the bank's overall credit-granting ability. In other words, financial conflicts lead to the discouragement of investment/lending of financially constrained banks. As a result, lenders have additional incentives to limit credit supply, even if they can afford to meet a specific need. In addition, the institutional characteristics (such as political stability, corruption, competition of the banking market) can significantly affect the relationship between accessing external credit and firms' operation ([22]). Therefore, it is necessary to control the interaction of institutional obstacles and firm characteristics in the relationship between exports and credit constraints.

This study focus on two analyses, including (i) Analysis 1: examining the effect of institutional obstacles on credit constraints; and (ii) Analysis 2: exploring the relationship between credit constraints and exports under institutional obstacles. To achieve these aims, this study adopts the cross-country data of the World Bank to focus on three research questions: (1) What is the effect of institutional obstacles on credit constraints? (2) What is the relationship between credit constraints and export? (3) How does the set of firm-specific characteristics and institutional obstacles affect the linkage between credit constraints and exports? Additionally, Beck2005 [15] found evidence that firms in less developed economies are more vulnerable to financial problems than those in other economic sectors. Additionally, liberalized financial markets and government support also reduces credit constraints for *SMEs* [23]. Therefore, to analyze the effect of credit constraints more clearly, this study adopts data in different countries classified regarding national income by the World Bank. The instrumental variable (*IV*) approach is applied to address endogenous problems of credit constraints. Moreover, since the dependent variables are binary, the models are estimated by *Probit* regressions. The robustness tests are conducted by using the substitute proxy for credit constraints and adapting to *SMEs* and large-sized firms.

This study makes several outstanding contributions. First, this study fills a gap in the literature on firm-level financial constraints. Previous studies analyzed financial constraints in many aspects, such as financial leverage, liquidity ratios, and overdrafts, but few studies focused on credit constraints. Significantly, the intertwined constraints between institutions, credit obstacles, and exports have not been considered. Unlike earlier studies, this study sheds light on the effect of institutional obstacles on credit constraints and scrutinizes its impact on the relationship between credit constraints and participating in the international market. Analyzing the interaction of institutions is necessary to perceive the social relationships and firm performance correctly. Second, there has been no research in the literature on the impact of institutional obstacles (including tax rate, political instability, corruption, and business licensing) on credit constraints. Especially, the study utilizes the World Bank firm-level data cross-countries. Furthermore, the paper provides a visualization of the marginal impact of institutional obstacles on credit constraints in two groups, including exporters and non-exporters. Therefore, our study might fill in this missing information. The importance of institutional constraints on firm performance is exposed. Based on that, the government can consider appropriate institutional reform strategies to encourage business development. Third, the remarkable contribution of the study is to cluster the data into four groups of countries based on their national income. Institutions and national growth appear to be linked [24]. Firms in the same group with typical institutional specifications might have similar responses to credit constraints and exports. Thus, grouping can bring more accurate results, thereby highlighting the role of institutions. As a result, the findings might provide ideas for managers and policymakers in specific areas. Through these, appropriate strategies for the unique characteristics of each economy can be designed. Fourth, this study briefly reviews the credit constraint measurements. A quantitative method through several criteria in the enterprise's financial statements is applied in the main analysis. This method is more thorough than others because it combines information about a firm's credit line, declined debt, and credit needs. At the same time, qualitative measurement methods are also approached in the robustness test section. Finally, this study conducts further sensitive analyses into the relationship between credit constraints and exports. In particular, each type of institutional constraint is observed separately. In addition, the paper carried out tests on two groups of enterprises classified by size.

The remainder of the paper discusses the literature review in Section 2. Then, Section 3 describes the empirical specification. In this section, the relevant data and estimation strategies are analyzed. Next, Section 4 presents all the results. Subsequently, Section 5 is the conclusion.

## 2. Literature Review

In the stream of the empirical literature, previous scholars approached a variety of indicators to represent credit constraints. Fazzari [5] was the first to develop the concept of a financially constrained firm based on the firm's dividend income ratio. The authors suggested that a low dividend–income ratio implies a higher investment cash flow sensitivity. Observations of financially constrained firms showed that they retained 94 percent of earnings and paid dividends in only 33 percent of the years. There were even many companies that had never paid dividends. These firms had a high average investment-to-capital ratio, and they used almost all of their cash flow for investment spending. However, Kaplan [25] found evidence that a high degree of investment sensitivity to cash flows is not related to a firm's financial constraints. The study approached the concept as follows: a firm faces financial constraint if the gap between the cost of internal capital and the cost of external capital increases—the more significant the gap, the higher the financial constraint of the firm. Similarly, Kadapakkam [26] examined this relationship in data from six *OECD* countries (Canada, France, Germany, Great Britain, Japan, and the United States). Contrary to initial expectations, as in the Fazzari study [5], the results showed that firms that are sensitive to investment cash flow are less likely to be financially constrained. Therefore,

using dividend payout ratio and cash flow sensitivity to measure financial constraints is controversial.

In addition, several studies used firms' financial ratios to proxy credit constraints/access to credit. These indicators are taken from the balance sheet of the firm. For instance, some indicators are used, such as liquidity ratio, leverage ratio, solvency, repaying ability, the cost of external financing [27], debt maturity, and long-term leverage [28]. However, some scholars suspected insufficient evidence for concluding that the low-level liquidity and low financial leverage imply a financial constraint [29–31]. A financially constrained firm may be inclined to store more cash in reserve for production. As a result, the firm can still secure liquidity but still be in a financial shortfall.

Similarly, financial leverage is often large in firms with higher liabilities than equity, implying that the firm has easy access to external finance. This risk can lead to negative assessments from lenders. As a result, this point might narrow the firm's ability to receive credit. Therefore, both the above indicators are better considered to measure the firm's financial health than financial constraints. Some other research, such as Love [32] and Wellalage [33], used overdraft facilities and loans from a financial institution to proxy credit constraints. However, according to this concept, credit-constrained firms include those who do not need to borrow capital and those who need to borrow but do not apply for a loan (because they can not satisfy the loan conditions). Meanwhile, the two concepts "*restricted*" and "*unnecessary*" are different in nature. In other words, a firm with no credit requirements is uncertain about whether or not it is credit-constrained.

Some other alternative measures commonly used in financial literature include the Kaplan–Zingales (*KZ*) Index [25], the Whited–Wu (*WW*) Index [34], and the Hadlock–Pierce (*HP*) Index [35]. In a brief review, the *KZ* Index is a relative measure of dependence on external financing. Firms with higher *KZ* Index scores are more likely to experience difficulties when financial conditions tighten because they may have trouble funding their ongoing operations. The *KZ* Index is built on five variables: cash flow-to-total capital (negative), the market-to-book ratio (positive), debt-to-total capital (positive), dividends-to-total capital (negative), and cash holdings-to-capital (negative):

*KZ Index* = −1.001909 · *Cash Flows*/*K* + 0.2826389 · *Q* + 3.139193 · *Debt*/*Total Capital* + −39.3678 · *Dividends*/*K* + −1.314759 · *Cash*/*K*;
*Cash Flows* = (Income Before Extraordinary Items + Total Depreciation and Amortization); *K* = total capital; *Q* = (Market Capitalization + Total Shareholder's Equity − Book Value of Common Equity − Deferred Tax Assets)/Total Shareholder's Equity; *Debt* = Total Long-Term Debt + Notes Payable + Current Portion of Long-Term Debt;
*Dividends* = Total Cash Dividends Paid (common and preferred); *Cash* = Cash and Short-Term Investments.

Similarly, the Whited–Wu Index measures financial constraints based on six variables: the ratio of the long-term debt to total assets, pays cash dividends, firm sales growth, the natural log of total assets, the firm's three-digit industry sales growth, and the ratio of liquid assets to total assets:

*Whited–Wu Index* = − 0.091 · *CF* − 0.062 · *DIVPOS* + 0.021 · *TLTD* − 0.044 · *LNTA* + 0.102 · *ISG* − 0.035 · *SG*;
*CF* is the ratio of cash flow to total assets; *DIVPOS* is an indicator that takes the value of one if the firm pays cash dividends; *TLTD* is the ratio of the long-term debt to total assets; *LNTA* is the natural log of total assets; ISG is the firm's three-digit industry sales growth; *SG* is firm sales growth.

By construction, firms with a high WW index are considered more financially constrained, characterized by low cash flow, low dividend, high leverage, low total assets, high industry sales growth, and low firm growth. The *HP* Index combines a company's asset size and age:

*HP − Index* = −0.737 · *Assets* + 0.043 · *Assets*$^2$ − 0.040 · *Age*;

*Assets* is the natural log of inflation-adjusted book assets and is capped at (the natural log of) USD 4.5 billion;
*Age* is the number of years a firm is listed with a non-missing stock price on Compustat and is capped at 37 years.

However, this index is not popular. In addition, a widely used method to measure the credit limit is to assess the firm's credit risk score. Credit institutions often use credit risk scores to decide the disbursement. In Muuls' study [1], the author used the Coface score as a measure of bankruptcy risk, so it is highly correlated with the level of credit constraints.

More comprehensively and rigorously, Kuntchev [36] introduced a very detailed credit constraints concept that considers the credit needs of firms and the level of credit provided. In particular, credit constraints were classified into four groups as Full Credit constraints (no access to loans), Partial credit constraints (provided part of credit needs by credit institutions), Maybe credit constraints (recorded in a line of credit/loan or overdraft facility), and No Credit Constraints. Later, Flaminiano [37], Su [3] also applied this method to measure credit constraints. To the author's knowledge, this is by far the best measure of the nature of credit constraints, and this method is widely applied.

In addition, owners' self-assessments about financial hardship are also used to represent credit constraints [38]. The perception scale of financial impediment ranges from zero (no obstacle for access to finance) to four (very severe obstruction for access to finance), considering access to credit as the biggest obstacle. Many studies have applied this measure in the financial literature [39]. However, this qualitative approach also contains certain limitations because these are non-objective assessments from interviewees [40].

The above measurement methods are very commonly applied in empirical studies related to the relationship between credit restrictions and exports. Based on the model of Melitz [10], this relationship had attracted the attention of many researchers and policy-makers. However, the results obtained are not consistent. Some studies even obtained conflicting results on the same observations group when examining the relationship in different years. Greenaway [41] was a pioneering study investigating the reverse causality relationship between financial constraints and export market participation. The study observed three groups of *UK* manufacturing firms, including continuous exporters, never exported, and starters. This separation allowed the study to reveal the impact of financial problems in detail. Specifically, the study found no significant difference in monetary terms between the non-exporting group and the start-exporting group. However, there was a discrepancy between exporting firms and new exporters. In addition, this research also demonstrated that small firms have more limited finances than large ones, and no clear evidence has been found that a firm with a financial advantage will become an exporter. Then, expanding the study of Greenaway [41], Bellone [30] surveyed 25,000 French firms in the period 1993–2005 to test the model with one-year lagged variables and up to three years lagged. This study highlighted differences in the firm's financial situation before exporting. The author calculated more than two other indicators (According to the method first published by Musso [42]) for measuring financial constraints. The results showed that exporters are firms with higher liquidity in the last year before exporting. In other words, financial constraints act as a barrier to participating in export. However, the Muuls model [1] showed that financial constraints positively affect export destinations but not the extensive export margin. The firm's productivity threshold for exporting was supposed to increase with the number of export destinations that a firm decides to serve. In other words, the more markets a firm exports to the higher its productivity should be and the fewer credit constraints it should have. Subsequently, empirical tests on 9000 Belgian manufacturing firms from 1999 to 2005 also supported the model. Financial constraints and export performance are closely related to export destinations and not to export turnover. Moreover, Bernard [39] observed firms from different industries in 28 Eastern Europe and Central Asia countries in 2005 and 2009. The authors found fascinating results about the causal relationship between credit constraints and exports. Specifically, the study found a positive effect of credit restrictions on exports in 2005 but no association between export

and access to finance in 2009. In addition, this study examined the responses of different firm groups by export status to changes in credit status. The results demonstrated that financially constrained firms in 2005 were more likely to become exporters in 2009. In addition, credit constraints did not affect the firm's decision to enter or exit the export market. However, comparing two financially constrained firms, a firm that exports continuously (exporting for two consecutive years) is more likely to improve its access to finance than a non-exporting firm that is financially constrained. After this, Qasim [31] also found no evidence that Pakistani firms can improve their financial constraints after they enter foreign markets in the short term. Due to the limited data set, his study only considered three years after joining export. Therefore, the author did not confirm this conclusion in the long term.

Notably, many scientists emphasized that political and institutional factors bind many economic and social phenomena. In other words, institutional specifications are likely to bias relationships in fields [43]. Therefore, controlling for institutional characteristics is the key to analyzing relationships realistically and deeply. The terminology "institutions" was first mentioned by Thorstein Veblen in 1898 [44], after which it has been widely applied to explain many behavioral phenomena and the decision-making processes of managers. Accordingly, the concept of "institution" includes regulations, principles, and laws used to regulate the activities of individuals and organizations. North [45] argued that the institutional framework consists of formal and informal institutions. Authorities implement official rules, including constitutions, statutes, charters, and legal documents. Meanwhile, informal rules can expand and govern subjects' behavior through social norms (traditions, beliefs, customs, taboos, etc.). Empirical researchers proposed several indicators to assess the institutions' quality, including political instability, corruption, political regime characteristics, social management (such as constitution, laws, procedures, and regulations), etc. A weak institution is characterized by political instability, severe corruption, suboptimal legality, and ineffective enforcement of regulations. On the contrary, it can be called a healthy institution.

Economists believe that the health of institutions can create an efficient investment environment. Firms might reduce transaction costs, increase investment activities and increase productivity. In particular, Rose-Ackerman [46] pointed out that difficulty obtaining export licenses and facing too many other complicated regulations might be detrimental to a firm's exportability. In addition, the cross-country empirical study of Dreher [47] discovered a negative association between entry regulation and the number of new entrants in the market. The cumbersome legal system with many excessive regulations will hinder the entry of new investors.

Banerjee [48] demonstrated the influence of institutional conditions on credit constraints in Indian companies. First, firms in agriculture, agro-processing, transportation, and small-scale industries (*SSI*) were the firms prioritized for credit by banks according to state regulations. Banks in this country only used 60% of the credit for non-priority industries. Consequently, the level of credit competition among firms in non-priority sectors became more intense. This fact made credit constraints in these industries likely to increase. Similarly, King [49], Levine [50], and Beck [51] showed that at the country level, the law and enforcement certainty directly influences the lending decisions , which in turn affects the ability of the firm to access the credit.

In brief, a weak institutional framework can affect access to finance. In addition, it is also a cause of limiting the export capacity of enterprises. Institutional specifications are likely to influence the relationship between credit constraints and exports.

## 3. Empirical Specification

### 3.1. Data Source

This study uses the latest firms survey provided by the World Bank and published in early 2021. The data set's inherent advantages are as follows: This rich data set covers more than 12 topics with over 100 indicators. This survey collects relatively complete information about the firm characteristics such as age, industry, location, number of employees, capital

status, export status, informal payments indicators, etc. These data are entirely consistent with the purpose of this study. Additionally, the study utilizes the World Bank's 2020 National Income Classification to categorize countries according to groups. Based on the Gross National Income (*GNI*) per capita of each country, the World Bank has divided the nations of the world into four income groups: Low-income countries (*LI*), Low-medium-income countries (*LMI*), Upper-medium-income countries (*UMI*), and high-income countries (*HI*). After cleaning the data and keeping the necessary indicators for the study, the remaining dataset contains about 137 thousand firms in 131 countries from 2010 to 2020 (see Table A1).

*3.2. Measuring Credit Constraints*

As mentioned in Section 2, there are two measurement credit constraints methods that have been commonly applied recently. They are a method through several credit criteria and the qualitative self-assessment method. We use both methods for this study. The quantitative measure for the main research and the qualitative approach for the robustness test.

First, we measure credit constraints via a firm's status on a credit line, credit refused, and demand for credit. This method follows Jappelli [13], Nguyen [14], and Su [3]. In particular: *(1) A line of credit (LoC)*: A firm is in a line of credit implies that it is in a credit arrangement with a formal financial institution. In other words, a firm that has a *LoC* means it needs a loan and has successfully approached a loan. In the survey, the information can be obtained from the question: *"At this time, does this establishment have a line of credit or loan from a financial institution?"*. This binary variable obtains a value of 1 if a firm has a line of credit or loan from a financial institution. Otherwise, it obtains a value of 0. *(2) Credit refused*: This indicator reflects that a firm has at least one denied loan application, which may imply that a firm is constrained in accessing credit even if it has demand. In the survey, the information can be obtained from the question: *"How many of those loan or line of credit applications were rejected?"*. A dummy variable (denoted as *Denied*) obtains a value 1 if a firm has at least one credit refused. Otherwise, it obtains a value of 0. *(3) Demand for a loan*: A notable highlight in this measurement method is the further control of borrowing demand. If having a *LoC* indicates that the firm has succeeded in accessing a loan. On the contrary, not having credit goes beyond the meaning of the firm not being able to access a loan. Because even though they have no credit restrictions, firms certainly will not be in a *LoC* if they do not need to apply for a loan (establishment has sufficient capital). Therefore, if loan demand is ignored, it may lead to deviations in determining credit constraint status. The information can be obtained from the question: *"What was the main reason why this establishment did not apply for a line of credit or loan?"* A dummy variable (named as *Demand*) obtains a value 0 if a firm does not apply for a loan because it does not have demand. Otherwise, it obtains a value of 1 if a firm does not apply for a loan because of another reason. The reasons might be (i) the procedure to apply for loans or credit lines is very complicated; (ii) interest rates are not favorable; (iii) collateral requirements for loans or credit lines are not achievable; or (iv) the enterprise does not think that the loan application will be approved and other reasons. Thus, the term "*constrained*" needs to be developed based on three factors, including loan demand, rejection, and *LoC*. Subsequently, a unconstrained firm has been granted all loan requirements or those who have not applied for credit because there is absolutely no need. In contrast, constrained firms (i) have been denied all credits or have only been partially accepted or (ii) need credit but cannot access a loan. The credit constraint variable (denoted as *CC*) is binary, taking a value equal to 1 if the firm has a credit constraint, and 0 otherwise. In particular:

$$CC = \begin{cases} 1 \text{ if a firm is in a line of credit and has at least one credit refused} \\ 1 \text{ if a firm doesn't apply for a loan but still has a demand} \\ 0 \text{ if a firm is in a line of credit but doesn't have any credit refused} \\ 0 \text{ if a firm doesn't apply for a loan because it does not have demand} \end{cases}$$

The second method to measure credit constraint is a qualitative method. In particular, the credit constraint is recognized by the firm's assessment instead of determining the

credit restriction through the firm's financial indicators. We spent the information recorded about the firm's perception of financial access. The specific question is: *"Is access to financing, which includes availability and cost [interest rates, fees, and collateral requirements] No Obstacle, a Minor Obstacle, a Moderate Obstacle, a Major Obstacle, or a Very Severe Obstacle to the current operations of this establishment?"* This standard proxy for credit constraints is applied in Bernard's study [39]. Although the subjective feelings of the business dominate this method, it overcomes the weakness of the binary variable in the first measurement. This determination method provides specific information about the severity of the credit restriction. By this method, the credit constraint variable is denoted as *FIN*:

$$FIN = \begin{cases} 0 \text{ if a firm has no obstacles in accessing finance} \\ 1 \text{ if a firm has a minor obstacle in accessing finance} \\ 2 \text{ if a firm has a moderate obstacle in accessing finance} \\ 3 \text{ if a firm has a major obstacle in accessing finance} \\ 4 \text{ if a firm has a very severe obstacle in accessing finance} \end{cases}$$

In this section, only formal credit (such as loans from banks, funds, credit organizations) are considered.

*3.3. Estimation Strategy*

Firstly, this study scrutinizes the effect of institutional obstacles on credit constraints. Secondly, the relationship between credit constraints and export participation under the moderation of institutional obstacles is estimated. Models are performed in four country groups including Low-Income Country (*LI*), Low-Medium-Income Country (*LMI*), Upper-Medium-Income Country (*UMI*), and High-Income Country (*HI*). Then, some further sensitivity analyses are conducted.

3.3.1. Analysis 1: The Effect of Institutional Obstacles on Credit Constraints

This analysis aims to determine the impact of institutional obstacles on credit constraints. In the baseline approach, we estimate the linear regressions of the following forms:

$$CC_{ijs} = \alpha_1 + \delta_1 \cdot Ins_{ijs} + v + \gamma + \epsilon_{ijs} \qquad \text{(Model 1)} \qquad (1)$$

$$CC_{ijs} = \alpha_2 + \delta_2 \cdot Ins_{ijs} + \sigma_2 \cdot Control_{ijs} + v + \gamma + \epsilon_{ijs} \qquad \text{(Model 2)} \qquad (2)$$

where subscripts $i$, $j$, and $s$ denote firm, country, and sector, respectively, and $v$ and $\gamma$ are year and sector fixed effects. $\epsilon_{ijs}$ is an error term.

In this study, we focus on a firm's perception of the obstacle of tax rates (denoted as *taxr*), business licensing and permits (abbreviated as *permit*), political instability (marked as *pol*), and corruption (represented as *cor*). They are four institutional obstacles that most firms considered to be the biggest obstacle. Firms self-assess the levels of obstacles that they face. Their perception of obstacles was assessed at the following levels: No obstacle, Minor obstacle, Moderate obstacles, Major obstacle, and Very severe obstacle. $Ins_{ijs} = tar, permit, pol, cor$ is the set of institutional obstacles.

$CC_{ijs}$ is a credit constraint measured quantitatively, as mentioned in Section 3.2. $CC_{ijs}$ is a binary variable that takes a value of one if a firm has a problem accessing credit. Otherwise, credit constraint takes a value of zero.

$Control_{ijs}$ the set of firm characteristics as a control variable (Control variables are similar to Phan2022 [52]). According to Beck [53], this paper controls some characteristics of a firm such as the natural logarithm of total assets (denoted as *fsize*) and the logarithm of firm age (marked as *fage*). In addition, firms located in big cities and urban areas can easily access bank capital for many reasons [54]. For example, there are more credit providers for their choice, and information between firms and credit providers is asymmetrical. Therefore, in this study, the firm's geographical location (marked as *loc*) is controlled and measured as the size of the locality (The size of locality is determined by population and classified in five groups: city with population over 1 million, over 250,000 to 1 million,

50,000 to 250,000, and less than 50,000). In addition, foreign ownership has the potential to influence a firm's decision to enter the international market [54] (According to Abor2008, companies with more than 10% foreign equity often have better access to information related to foreign markets. Ownership of more than 10% in companies that demonstrate the right to express an opinion at the general meeting of shareholders can influence decisions in the company's operations [52]). Consequently, this study creates a dummy variable (marked as *own*) obtains the value 1 if the firm is owned 10% or more by foreign individuals, companies, or organizations, 0 if otherwise. Additionally, Flaminiano [37] suggested that a firm with innovative activities (including technological innovation, product improvement, and new product development) creates positive points that make lenders believe in the ability of a project to succeed and a firm's debt repayment capacity. Consequently, it may be necessary to consider innovation (marked as *innov*) and international certification (represented as *ctfc*) as control variables. We also control the logarithm years of experience working as the top manager in a firm of a manager (named *mae*) [17]. Finally, the export status (denoted as *exp*) is controlled as a dummy variable. All symbols and subscripts are unchanged throughout the remainder of this study.

Then, we report the predicted marginal effects of four institutional obstacles in four country groups on credit constraints at the 95% confidence interval. We compare two groups by export status.

In the robustness test, instead of quantitative credit constraints (*CC*), we approach qualitative credit constraints (denoted as *FIN*) (which mentioned in Section 3.2). We use the information recorded about the firm's perception of financial access. The specific question is: *"Is access to financing, which includes availability and cost [interest rates, fees, and collateral requirements] No Obstacle, a Minor Obstacle, a Moderate Obstacle, a Major Obstacle, or a Very Severe Obstacle to the current operations of this establishment?"*.

According to Wooldridge [55], the *Probit* model is suitable to estimate the probability of credit constraint. All the average margins effect is reported.

### 3.3.2. Analysis 2: The Linkage between Credit Constraints and Export Decision

Numerous empirical studies pointed out the importance of finance for a firm's development. In this study, we sought to distinguish whether the results were the same in economies with different wealth levels. Model 3 is regressed to interpret the impact of credit constraint on export as follows:

$$ExD_{ijs} = \alpha_3 + \beta_3 \cdot CC_{ijs} + \sigma_3 \cdot Control_{ijs} + v + \gamma + \epsilon_{ijs} \quad \text{(Model 3)} \quad (3)$$

where subscript *i,j,s* denote firm, country, and sector, respectively. $v$ and $\gamma$ are year and sector fixed effects. $CC_{ijs}$ represent measured credit constraint of firm *i*, while $Control_{ijs}$ is the set of control variables, which is the same as in Model 1 and 2. $\epsilon_{ijs}$ is an error term. In this study, only the direct export is considered as a dummy variable (denoted as *ExD*).

Then the influence of credit constraint on the export is scrutinized by adding institutional obstacle variables into Model 4, as follows:

$$ExD_{ijs} = \alpha_4 + \beta_4 \cdot CC_{ij}s + \delta_4 \cdot Ins_{ijs} + \sigma_4 \cdot Control_{ijs} + v + \gamma + \epsilon_{ijs} \quad \text{(Model 4)} \quad (4)$$

where $Ins_{ijs} = taxr, pol, cor, permit$ is the set of institutional obstacles (similar to Models 1 and 2). Theoretically, many researchers assumed that a causal relationship might exist between credit constraint and export, such as [31,39]. First, a firm with less credit constraint is expected to increase its ability to enter export markets. Conversely, exporting might improve a firm's access to external capital. Due to this concurrent relationship, the variable of credit constraint might be endogenous. In addition, credit constraint is correlated with unobserved factors. Although firm characteristics are controlled in the model that determines the correlation between credit restriction and export decision, some unobserved features still exist. They are, for example, the implicit relationship between the company and the credit institution; information asymmetry; factors related to the risk of the loan such

as loan purpose and maturity time; and information beyond corporate reports that lenders collect through their network [3]. In addition, the credit constraint is measured based on self-reported data from the firm, which may be misleading due to biased judgments or obscured information [1]. These signs showed that the credit constraint variable is endogenous in the model. Therefore, using *OLS* regression can lead to bias. It is necessary to choose the instrumental variable [55].

According to Gatti [56], inspection is the driving force for firms to pay more attention to regulatory compliance. The audit results ensure the reliability of the Profit and Loss Statement and other regarding a firm's compliance. Consequently, if the inspection results are promising, these firms have more opportunities to access credit. Conversely, an inspection can yield negative consequences for access to credit if the results are judged to be disqualifying. Thus, government inspections are more likely to affect firms' access to credit [57]. Therefore, we use information about corporate audits as an instrumental variable. This information is obtained from the from survey question: *"In the fiscal year [insert last complete fiscal year], did this establishment have its annual financial statements checked and certified by an external financial auditor?"*. From there, the dummy instrumental variable is created (abbreviated as *audit*), which takes the value 1 if the answer is Yes and the value 0 if the answer is No.

Since the exported variable is binary, the *IV-Probit regression* method is applied. In addition, we use $v$ and $\gamma$ as year and sector fixed effects. In all regressions, the average marginal effect is reported.

In the next step, each type of institutional obstacle is examined individually. We separate each institutional obstacle into two groups. One group consists of firms that face an institutional obstacle (at minor, moderate, major, and very severe levels). Another group includes no institutional obstacle firms. Since then, Model 3 is regressed in sub-samples of firms. The comparison regarding facing obstacles shows whether the impact of credit constraints on exports varies across different constraint groups. Then, it is possible to assess the moderating effect of each institutional obstacle on this relationship. The set of control variables are similar, and the regressions are performed on four sub-data classified by country income group.

According to Burkart [58], firm size is an essential factor that plays a decisive role in firms' capital structure and business decisions. Large firms are trusted to have many qualities to guarantee loan repayments, such as high reliability, long-standing relationships with banks, and diversified collateral. Meanwhile, small firms are often characterized by limited information resources related to lenders and a lack of collateral [59]. Therefore, despite their significant contribution to the economy, *SMEs* face more difficulties accessing credit. In the light of economic integration becoming a trend, *SMEs* are presented with many opportunities to expand and develop foreign markets. After the opening of trade, the need for external finance for these enterprises becomes even more significant [60]. *SMEs* have a sharp increase in capital needs to offset the sunk costs of exports, improve product quality, train workers' skills, etc. Therefore, easy access to credit is likely to facilitate firms participating in international markets. Recognizing the essential role of firm size, we use regression Model 3 and 4 on two sub-samples, including *SMEs* and Large-sized firms in the further analysis.

## 4. Estimation Results and Discussion

### 4.1. Analysis 1: The Effect of Institutional Obstacles on Credit Constraints

First, Table 1 presents the influence of institutional obstacles on credit constraints in four country groups. Columns (1), (3), (5), and (7) are the results of regressions without the set of control variables (Model 1), and Columns (2), (4), (6), and (8) are the results of Model 2.

**Table 1.** The influence of obstacles on credit constraint (*CC*) in four country groups.

| | LI | | LMI | | UMI | | HI | |
|---|---|---|---|---|---|---|---|---|
| | Model 1 (1) | Model 2 (2) | Model 1 (3) | Model 2 (4) | Model 1 (5) | Model 2 (6) | Model 1 (7) | Model 2 (8) |
| taxr | 0.050 *** | 0.044 ** | 0.099 *** | 0.111 *** | 0.061 *** | 0.124 *** | 0.087 *** | 0.097 *** |
| | (0.011) | (0.017) | (0.006) | (0.007) | (0.007) | (0.011) | (0.014) | (0.019) |
| pol | 0.030 *** | −0.008 | −0.021 *** | −0.013 * | 0.019 ** | 0.027 ** | 0.048 *** | 0.068 *** |
| | (0.011) | (0.017) | (0.005) | (0.007) | (0.007) | (0.011) | (0.015) | (0.021) |
| cor | 0.016 | −0.015 | 0.043 *** | 0.048 *** | 0.019 ** | 0.013 * | 0.093 *** | 0.079 *** |
| | (0.011) | (0.017) | (0.006) | (0.007) | (0.007) | (0.011) | (0.014) | (0.020) |
| permit | 0.055 *** | 0.048 ** | 0.047 *** | 0.047 *** | −0.010 | −0.015 | 0.060 *** | 0.079 *** |
| | (0.012) | (0.018) | (0.006) | (0.008) | (0.008) | (0.012) | (0.015) | (0.021) |
| exp | | −0.067 | | −0.096 *** | | −0.084 ** | | −0.127 ** |
| | | (0.080) | | (0.028) | | (0.038) | | (0.059) |
| fage | | −0.023 | | −0.045 *** | | −0.062 *** | | −0.054 * |
| | | (0.027) | | (0.012) | | (0.018) | | (0.032) |
| fsize | | −0.202 *** | | −0.113 *** | | −0.127 *** | | −0.034 * |
| | | (0.019) | | (0.007) | | (0.011) | | (0.019) |
| loc | | −0.073 *** | | 0.059 *** | | −0.015 | | −0.052 *** |
| | | (0.022) | | (0.0079) | | (0.011) | | (0.020) |
| ctfc | | −0.233 *** | | −0.054 ** | | 0.019 | | 0.106 ** |
| | | (0.068) | | (0.023) | | (0.034) | | (0.050) |
| mae | | 0.018 | | 0.006 | | −0.005 | | −0.034 |
| | | (0.029) | | (0.012) | | (0.018) | | (0.033) |
| innov | | −0.001 | | 0.109 *** | | 0.007 | | 0.041 |
| | | (0.041) | | (0.019) | | (0.030) | | (0.056) |
| own | | −0.002 | | −0.050 | | −0.142 *** | | −0.074 |
| | | (0.056) | | (0.033) | | (0.052) | | (0.073) |
| Const | −0.308 *** | 0.605 *** | −0.961 *** | −0.732 *** | −1.017 *** | −0.487 *** | −1.470 *** | −0.873 *** |
| | (0.071) | (0.137) | (0.034) | (0.059) | (0.046) | (0.082) | (0.124) | (0.196) |
| Obs | 9660 | 4596 | 35,212 | 24,415 | 26,364 | 12,967 | 9042 | 5423 |

Note: Standard errors in parentheses. ***, **, * denote significance at 1%, 5%, 10%, respectively. All regressions include the year and sector fixed effects. LI, LMI, UMI, and HI are four country groups regarding the national income.

Overall, most results show that all four institutional obstacles positively correlate with credit constraints in Models 1 and 2, except for a few special cases. These findings imply that firms are likely to increase their credit constraint as pressures from obstacles increase, such that excessive tax rates, business license administration in government agencies, political instability, and corruption are exacerbated. In particular:

*Tax rate:* Tax rate constraints exacerbate credit problems in all countries. The correlation coefficients of tax rates are all positive and significant at the 1% level, regardless of the presence of the control variable in the models (except column 2, significance level 5%). Comparing the four types of institutional obstacles, the tax rate barrier has the most substantial impact on increasing the probability of credit restriction, except for the *LI* group (Columns 1 and 2). Meanwhile, comparing between countries, the unreasonable tax rate has the most potent effect on credit status in firms of the group *LMI* (Columns 3, 4). This result is consistent with the inferences about the role of state control over business in developing countries. Fisman [61] and Johnson [62] pointed out that firms can receive advantages such as tax breaks, credit subsidies, etc., based on political connections. [63] Kim's research showed the impact of the relationship between government and firms on corporate tax rates in Southeast Asian companies. Meanwhile, Adhikari [64] and Derashid [65] also found

evidence that tight government control over the firm is more robust in developing countries. Therefore, firms in these countries are more likely to take advantage of political connections to receive tax favors than in developed countries. In addition, the tax costs of firms are diverse and expensive. In addition to corporate income tax, their other business activities are also subject to national tax (e.g., capital transfer tax, foreign contractor tax, commercial property tax). Tax costs account for more than one-third of total expenses (according to Worldbank's assessment (https://www.doingbusiness.org/en/reports/thematic-reports/paying-taxes-2020 (accessed on 5 April 2022)). Therefore, increasing tax rates is likely to reduce the attractiveness of external credits to firms. Its impact might be more potent than other types of institutional obstacles. Countries often use tax rates as an effective tool to gain strategic goals. The government might impose different tax rates on business-oriented groups. For example, Vietnam applies a standard corporate tax rate of 20%. However, in some special cases (such as the oil and gas industry and mining, depending on each project's location and specific conditions (Article 11 Circular 78/2014/TT-BTC, Law on corporate tax in Vietnam)), the corporate tax rate is 30 to 50%. To encourage new entry firms, firms operating in disadvantaged areas, and those employing many female workers, the government applies a regime that allows taxpayers to enjoy tax incentives or tax reductions. A high tax rate can hinder and even stop a firm's operation.

*Political instability*: Political instability almost negatively affects the ability of firms to access credit in groups of countries. In Columns 1 and 5–8, an increase in political instability might be detrimental to firms' access to finance, as the correlation is positive. Once the political environment becomes prone to instability, it will be the most significant disadvantage for firms in the developed economy (*HI* group) in accessing finance. This conclusion supports the view of Roe [66]. She argued that political factors drive financial market development. A stable, democratic politics can ensure the interests of investors and protect assets for firms. Therefore, political instability creates an inefficient investment environment and many uncertainties in business. In addition, due to the risk of political instability, banks might restrict disbursement to high-risk industries by tightening loan conditions, increasing collateral conditions, etc. As a result, firms' opportunities to access loans are narrowed [67]. More interestingly, we find a negative relationship between political instability and credit constraints in the *LMI* countries (Columns 3, 4), but the effect is the opposite for the rest. This result implies that instability in countries *LMI* is likely to reduce corporate credit constraints. Dinc [68] explained that state-owned banks are likely to loosen conditions and increase corporate lending during the election process. In addition, Barro [69] commented that political instability covers many aspects, such as instability in the government apparatus, political violence, frequent government changes, policies, etc. Therefore, the relationship between political instability may be different if the causes separate the types of instability. In this study, we did not clearly distinguish the types of instability. Despite government ownership being likely to affect the access to capital of firms [68], the ownership form of credit institutions is also not considered in this study. Therefore, although we cannot provide a definite explanation for the phenomenon in *LMI* countries, the results give some confidence in their interactions, similar to the conclusions of Roe [66].

*Corruption*: Corruption is not only a leading problem in developing countries or emerging economies. This problem is global and manifests in many different forms. Corruption seriously affects the sustainable development of the economy and society [70]. In particular, some researchers believe that corruption reduces competitiveness and distorts the nature of trade [71]. Therefore, a country with a high rate of corruption is often unattractive to foreign investors and domestic investors. As a result, the domestic economy cannot afford to face difficulties caused by integration and free trade. Moreover, according to Olken [72], the manipulation of bribe-takers in the state management apparatus makes the rights and interests of business entities infringed. Over time, it weakens a country's institutional system. However, this research did not find out a relationship between corruption and credit constraints in the *LI* group (Columns 1 and 2) (as the correlation coefficients are

significant at a level of more than 10%). Corruption is likely to increase by 4.8% (Column 4) the probability of credit constraints in the *LMI* group. For the model without control variables, the impact of corruption on credit constraints is less than insignificant (0.3%, column 3). This finding aligns with Wellalage' study [73] that in five South Asia countries including Afghanistan, Bangladesh, India, Nepal, and Pakistan in 2014 (According to the World Bank's 2020 National Income Classification, Afghanistan is *LI* country, the remaining four countries are *LMI* countries Appendix A. However, the number of observed firms in Afghanistan only account for 3.77%. Thus, these findings can represent the *LMI* group). The results showed that corruption increases the probability of credit constraints by almost 8%. The impact of corruption was most significant in the *HI* group. The probability of a credit constraints can increase by 9.3% (Column 8) when a business has a corruption problem. This is almost five times higher than the impact of this barrier in the *UMI* group (Column 5—correlation coefficient 0.019). These findings are consistent with some previous views, such as Meon [74] and Shleifer [75]. The authors supported the view that corruption poses a major disadvantage to business development, especially access to finance. In addition, our findings are consistent with the hypothesis of Avnimelech [76]. Based on Huntington [77] and Leff [78], Avnimelech argued that corruption increases the cost of borrowing and creates a negative assessment of the stable development of the market. Therefore, it negatively impacts firms' need to enter and expand the market. This conclusion implies that firms may experience a reduced need for loans due to psychological effects caused by corruption, even though they still have demand. In addition, Avnimelech found evidence that doing business in developed countries is particularly sensitive to corruption. The negative impact of corruption in developed countries is twice that of non-developed countries. Therefore, our results imply that anti-corruption is essential for every country. However, high-income countries should have stricter measures focused on reducing the negative effects of corruption.

*Business Licensing and Permit*: Our study finds evidence about the effect of a business licensing on accessing external credit, except for firms in the *UMI* group (Columns 5 and 6). Columns 2 and 4 show that when firm characteristics are controlled, business licensing obstacles are likely to increase the probability of credit constraints in *LI* and *LMI* groups, 4.7% and 4.8%, respectively. These findings are consistent with several previous studies. Shleifer [79] demonstrated that the law, regulations, and conditions are all critical to the size and extent of capital markets. Therefore, the enforcement of laws and administrative procedures can affect the willingness to lend and the ability to access credit. Similarly, Skosples [80] emphasized that the regulations about establishment and closure in transition economies negatively influence bank lending decisions. The author argued that the overlapping regulations and the legal inefficiency directly affect the disposal of bank collateral if the loan is not reimbursed. In the *HI* group, the impact of this obstacle is more significant. In particular, the probability of credit constraints is likely to increase by 6% (Column 7) when firms face difficulties with a business license and permit. However, differences in firm and owner characteristics emerge (Column 8), suggesting that firms find it harder to access credit when they face this barrier, as the probability of credit constraints increases by 7.9%.

Although the results indicate that if the government introduces strict and complicated procedures for licensing, it can be detrimental to firms in accessing formal loans. To increase their chances of receiving healthy credit, restructuring the legal process, removing cumbersome regulations, and strengthening the legal framework are necessary. However, this recommendation does not mean that all barriers need to be removed. Although it can create favorable conditions for firms, removing barriers poses many risks to credit institutions and the economy. For example, minimum capital provisions for the issuance of business licenses, export licenses, certificates related to collateral, origin, etc., are necessary to guarantee loan repayment. Therefore, a reasonable regulatory framework and proper licensing procedures are needed rather than repealed.

Moreover, the negative correlation coefficient of control variables also supports the previous hypothesis. The results show that it is easier for exporting firms to obtain loans than domestic ones. In addition, firm size is also correlated with access to credit. In addition, the significant negative relationship between a firm's age and credit restriction implies that long-standing firms have more advantages than young ones regarding access to credit. Beck [53] determined that older and larger firms are associated with better management capacity, more collateral choices, and high reputation. These are positive factors that determine the credit-worthiness of the firm. Therefore, large and older firms are more likely to access credit than young and small-scale firms [81,82].

In addition, possessing international certifications means that the firm is recognized for quality and reputation. Therefore, international certification can create favorable conditions for firms to access external funding. Nevertheless, the coefficient between innovation, foreign ownership, and credit constraint is insignificant ($p > 0.1$). The results indicate that firms located in large cities of middle-income countries (*LI* and *HI*) might access credit more easily. This finding is the opposite of firms in *LMI* countries. As pointed out by Lee [83], geographic location significantly affects credit status. The author believed that the city's geographical size is associated with economic advantages. He surveyed 97 countries and found that large cities—more than 1 million inhabitants—facilitate access to finance better than other cities because firms located in the big city receive more opportunities to access advanced facilities, increase network connectivity, and increase professional training [84]. Some scientists, such as Brambor [85] and Martin [86], argued that the geographical gap would gradually disappear due to the growing digital science. However, the empirical results show that the influence of geographical location on the development of financial markets and firms cannot be denied, especially in developed countries.

In brief, all four mentioned institutional obstacles affect the firm's credit status in the *HI* and *LMI* groups. Meanwhile, corruption in the least developed countries (the low-income nation—*LI*) does not seem to correlate with access to credit. In addition, the excessive control of administrative procedures such as the business licenses and permits aggravates the firm's limited credit situation, except for the *UMI* group. Finally, the impact of political stability is different across country groups.

In the second step, we predict the margin effect of the obstacles levels for credit constraints. Then, we compare non-exporters and exporters to further assess the impact of each perceived level obstacle on credit constraint. Figure 1 provides a visual representation of the marginal effect of obstacles on credit constraints at the 95% confidence interval, compared between four groups of countries. In general, firms' perceptions of barriers have a significant marginal effect on credit constraints. At first glance, the charts of the two groups by export status do not reflect a stark difference. The results found heterogeneity in the trend of the impact of constraints on credit restriction in the *LI* , *LMI* , and *UMI* groups. However, the image of the *HI* group showed a similar effect of the hindrances. An exciting feature is that in the first two groups (Figure 1(1–4)), the impacts of all types of obstacles on non-exporters across all levels are significantly higher than those of exporters. Meanwhile, there is little difference between the two groups by export in the rest of the countries (Figure 1(5–8)). Numerous previous empirical studies scrutinized a correlation between institutional quality and economic growth. Easterly [24] argues that underdeveloped countries in Africa are characterized by weak institutional frameworks, poverty, and backwardness. In addition, Hall [87] and Easterly [88] found that countries with better institutions are often associated with high *GDP* per capita. This impact mechanism is explained through the growth of *FDI* investment and the speed of opening of the economy. The above arguments imply that efficient institutions often characterize developed countries. Therefore, discrimination between firm groups is narrowed. This hypothesis suggests that institutional obstacles to credit constraints are nearly the same in all firm groups, regardless of export status.

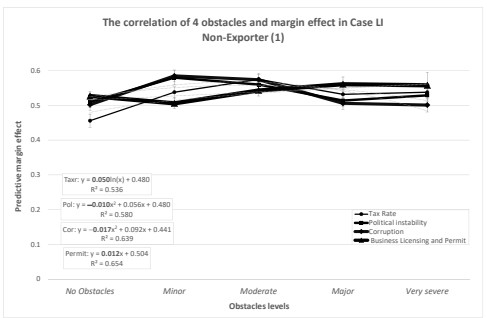

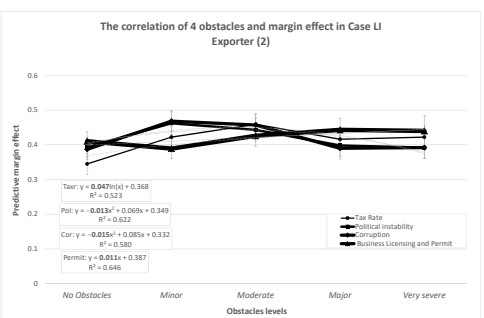

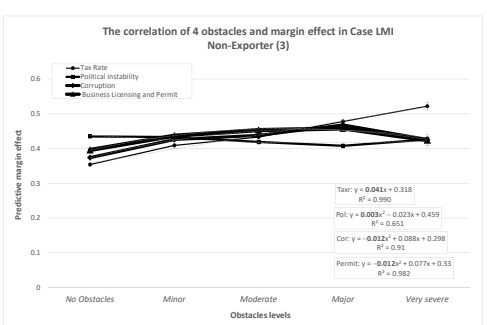

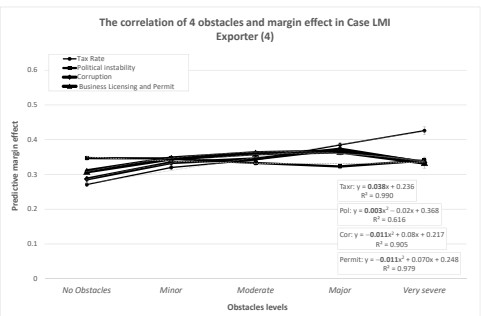

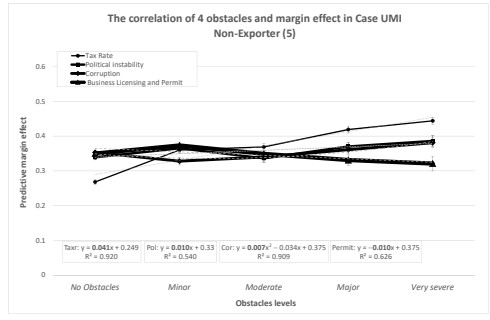

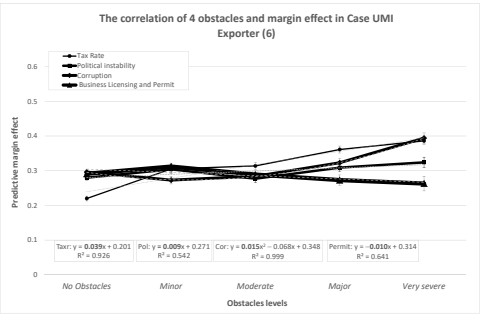

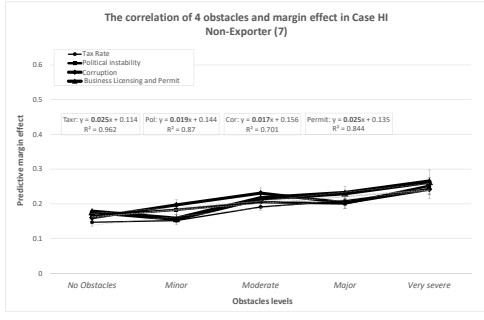

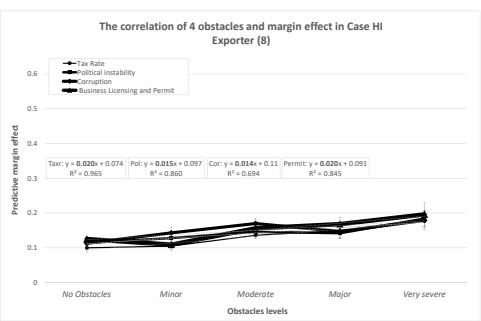

**Figure 1.** Visualization of the predicted marginal effects at the 95% confidence interval. Subfigures 1 and 2 are the *LI* group, subfigures 3 and 4 are the *LMI* group, subfigures 5 and 6 are the *UMI* group, and subfigures 7 and 8 are the *HI* group.

In more detail, the relationship between tax rate and the probability of the credit constraint is predicted as a positive linear function, with $R^2$ being more than 92%, except in the *LI* group. Access to credit becomes increasingly disadvantageous as tax barriers become more severe for firms. The results show that the marginal effect of tax rates is most substantial in medium-developed countries (*LMI* and *UMI* groups) and lowest in developed countries (*HI* groups). The proof is that the histogram increases steadily with a smaller amplitude in the *HI* group, compared to the other groups. Although the tax rate impact is predicted to be a saturation function in *LI* countries (Figure 1(1,2)), the coefficient of determination is only about 50%. This value of $R^2$ might suggest that the tax rate association explains only about 50% of the differences in credit restriction between individuals. This implication can lead to doubts about the appropriateness of the model. However, after testing the model prediction with other functional forms such as linear, polynomial, etc., the saturation function seems most suitable. The trendlines are predicted as polynomial functions for the remaining three barriers. The cubic function is predicted to be suitable for the group of emerging countries (*LI*), and the polynomial of order 2 is ideal for the group of *LMI*.

In contrast, in more prosperous countries (*UMI* and *HI*), the majority conforms to the linear functional form (except for corruption in *UMI* countries). In the *LI* countries, the impact of political instability and corruption on the margin effect of credit constraints is similar for both exporting and non-exporting groups (Figure 1(1,2)). The most explicit disparity in the marginal impact of credit constraints is between the No Obstacles and Minor levels. The marginal effect of credit constraints increased sharply before trending down at subsequent levels of barriers. The result then tends to become saturated.

The marginal effect of credit constraints decreases once firms move from a perception of no obstacles to a perception of a mild impediment to business licensing restrictions. After that, this marginal effect tends to increase significantly before saturating at the Major level. For the (*LMI* group of countries (Figure 1(3,4)), the predicted trend lines are mainly polynomials of order two, where corruption and permit are concave functions. Higher levels of national corruption and tight policy regulation increase the expected marginal and peak near 0.5 when the barrier is Major level.

In *UMI* countries (Figure 1(5,6)), the marginal effect of corruption is predicted to be a convex quadratic polynomial. As the level of corruption gradually moves from low corruption (Minor) to higher levels, it increases the margin effect on credit restriction. However, the graph is steeper for exporters (Figure 1(6)) compared to domestic firms (Figure 1(5)). Instead of fluctuating as much as other groups, in the prosperous country group (*HI*), institutional weakness affects credit constraints and is predicted as a linear function.

Thus, it can be seen that under the influence of different country characteristics (different average income), the impact of some unfavorable institutional features has a different effect on accessing credit. Hence, the graphs showing the marginal impact in each country group can be described by different functional forms. The graphs fluctuate more strongly in poorer countries. These results suggest that policymakers in emerging countries might clean up the morality of the government apparatus, restructure legal processes, etc., to limit the adverse effects of a weak institutional system. The government should offer more tax support policies (tax reduction, tax return support, and stricter tax management) to create favorable conditions for firms' development.

In the third step, Table 2 summarizes the regression results in the robustness test. The credit constraints is measured by a qualitative method. Overall, all regression coefficients are statistically significant at the 1% level. In addition, all four institutional obstacles have a positive relationship with credit constraints. In other words, an increase in the perception of institutional obstacles can increase the likelihood of perceived financial limitations. In addition, regardless of controlling for firm-specific variables, the results did not change significantly. These findings support the main regression results (shown in Table 1).

**Table 2.** Results of alternative credit constraints (*FIN*).

| | LI | | LMI | | UMI | | HI | |
|---|---|---|---|---|---|---|---|---|
| | Model 1 (1) | Model 2 (2) | Model 1 (3) | Model 2 (4) | Model 1 (5) | Model 2 (6) | Model 1 (7) | Model 2 (8) |
| taxr | 0.100 *** (0.0102) | 0.063 *** (0.010) | 0.145 *** (0.005) | 0.147 *** (0.001) | 0.182 *** (0.005) | 0.204 *** (0.007) | 0.172 *** (0.008) | 0.177 *** (0.011) |
| pol | 0.039 *** (0.010) | 0.040 ** (0.016) | 0.082 *** (0.005) | 0.092 *** (0.006) | 0.068 *** (0.005) | 0.102 *** (0.007) | 0.055 *** (0.009) | 0.073 *** (0.012) |
| cor | 0.069 *** (0.010) | 0.011 (0.016) | 0.039 *** (0.005) | 0.047 *** (0.006) | 0.085 *** (0.005) | 0.092 *** (0.008) | 0.119 *** (0.009) | 0.136 *** (0.012) |
| permit | 0.230 *** (0.012) | 0.222 *** (0.019) | 0.209 *** (0.006) | 0.208 *** (0.007) | 0.164 *** (0.006) | 0.158 *** (0.009) | 0.210 *** (0.009) | 0.230 *** (0.013) |
| exp | | 0.108 * (0.065) | | −0.005 (0.022) | | −0.029 (0.024) | | 0.043 (0.031) |
| fage | | −0.006 (0.025) | | −0.021 ** (0.010) | | −0.074 *** (0.013) | | −0.089 *** (0.018) |
| fsize | | −0.082 *** (0.01) | | −0.080 *** (0.006) | | −0.021 *** (0.001) | | −0.006 (0.011) |
| loc | | −0.057 *** (0.021) | | 0.005 (0.006) | | 0.021 *** (0.008) | | −0.032 ** (0.012) |
| ctfc | | −0.173 *** (0.058) | | −0.082 *** (0.018) | | 0.015 (0.022) | | 0.027 (0.029) |
| mae | | −0.048 * (0.026) | | −0.032 *** (0.011) | | −0.019 (0.013) | | −0.073 *** (0.019) |
| innov | | 0.009 (0.039) | | 0.184 *** (0.016) | | 0.070 *** (0.020) | | −0.073 ** (0.030) |
| own | | −0.126 ** (0.051) | | −0.062 ** (0.027) | | −0.183 *** (0.033) | | −0.204 *** (0.041) |
| Const | −0.259 *** (0.058) | 0.379 *** (0.122) | −0.164 *** (0.030) | 0.199 *** (0.051) | −0.310 *** (0.036) | −0.140 ** (0.062) | −0.804 *** (0.054) | −0.204 * (0.106) |
| Obs | 13,653 | 6678 | 53,794 | 37,170 | 50,282 | 26,343 | 19,465 | 11,999 |

Note: Standard errors in parentheses. ***, **, * denote significance at 1%, 5%, 10%, respectively. All regressions include the year and sector-fixed effects. LI, LMI, UMI, and HI are four country groups regarding the national income.

However, the level of institutional impact on firms' access to finance is stronger than the quantitative credit constraints. For example, in *LI* countries—Column 1—the marginal effect of the business licensing barrier on the alternative credit constraint is 0.230, which is four times higher than on the original credit constraint (Column 1, Table 1). In addition, according to these qualitative credit constraints, the impact of licensing barriers is more pronounced. This barrier has the most substantial influence compared with the remaining obstacles (except for the *UMI* group). In addition, the effect of control variables is not different from the results in Table 1.

In conclusion, access to credit is governed by an institutional framework in every country. A weak institutions inhibits firms' opportunities to access finance. Furthermore, the firm's characteristic contributes to this negative. These results suggest that developing countries should build a more affordable tax system for small firms. Additionally, governments need to maintain political stability and clean up bureaucracy. Importantly, this is not only necessary for poor and emerging countries. The effects of political instability and corruption are also evident in prosperous economies. The cumbersome and redundant legal procedures are barriers to the development of enterprises. Therefore, restructuring the management and licensing mechanism is essential to support enterprises in accessing capi-

tal. Finally, the establishment of mechanisms and policies to help *SMEs* and firms in remote areas are also measures to consider to reduce their difficulty in accessing financial resources.

### 4.2. Analysis 2: The Linkage between Credit Constraints and Export Decision

The endogenous tests is performed and presented in Table 3. The Hausman test results reveal that $\chi^2 = 26.021$ and is significant with $\rho = 0.000$. This result confirms the existence of the endogenous phenomenon and, at the same time, shows that the selected instrumental variable meets the requirements that there is no correlation with the residuals of the original regression model. Therefore, it has successfully overcome the endogenous phenomenon caused by the credit constraint variable. This conclusion is strongly supported by the results of the LM-statistic and the Cragg–Donald–Wald *F*-statistic because the results of these two tests are significant with $\rho = 0.000$. In summary, these tests provide evidence to confirm that the instrumental variable is valid and has adequate power in mitigating the endogeneity problem.

**Table 3.** Test for the endogenous variable.

|  | **Coefficient** | **$\rho$-Value** |
|---|---|---|
| Hausman test of endogeneity ($\chi^2$) | 26.021 | 0.0000 |
| Anderson Canon.Corr.LM statistic (Under identification test) | 134.794 | 0.0000 |
| Cragg–Donald–Wald F-statistic | 135.078 | 0.0000 |

Note: Endogeneity Test is constructed by using 2SLS regression.

Then, Table 4 summarizes the effects of credit constraints and control variables on exports with and without controlling institutional obstacles. Overall, the findings show a negative impact of credit constraints on firms' exports to international markets. However, these adverse effects tend to increase with the wealth of countries. Looking at the group of lowest-income countries (Column 1), we find no relationship between credit constraints and export status. Weak institutions can increase uncertainty and the cost of trade, leading to underperforming markets [89]. Consequently, firms in poor and emerging countries are more dependent on institutional quality [90]. This robust dependence might be why removing institutional obstacles from the model is inappropriate. After further examination of institutional specifications, Column 2 shows the negative outcome of credit constraints. Under institutional obstacles, firms that struggle to access credit might reduce their ability to participate in exports by 4.6%. These firms seem to be affected only by tax rates and licensing procedures. The interaction between exports and the other two types of institutions, including political instability and corruption, was not statistically significant. This result evokes contemplation because, according to Olken [72], corruption and instability are severe problems in underdeveloped and developing countries.

On the other hand, North [45] argued that the impact of informal institutions is more significant in weak economies. In addition, informal funding is the most common for small-size firms. However, in the framework of this study, only formal credits are considered. This is why the results in this group were not as expected.

Despite the institutional obstacles, credit constraints presumably affect their ability to participate in international markets in the remaining three groups. This findings align with Muuls [1], Manova [91]. Under institutional interaction, credit-restricted firms are likely to reduce the probability of firms exporting in *LMI* and *UMI* countries by 8.1% (Column 4) and 6.6% (Column 6), respectively. These negative effects were more severe in the *HI* group. In particular, the probability of exporting is likely to be reduced by approximately 20% (Column 8) when enterprises have difficulty accessing capital.

**Table 4.** The influence of credit constraint (*CC*) on exports in four country groups.

| | LI | | LMI | | UMI | | HI | |
| --- | --- | --- | --- | --- | --- | --- | --- | --- |
| | Model 3 (1) | Model 4 (2) | Model 3 (3) | Model 4 (4) | Model 3 (5) | Model 4 (6) | Model 3 (7) | Model 4 (8) |
| CC | −0.038 (0.062) | −0.046 * (0.304) | −0.067 *** (0.024) | −0.081 *** (0.024) | −0.55 ** (0.028) | −0.066 *** (0.036) | −0.244 *** (0.056) | −0.200 *** (0.651) |
| Tax | | 0.055 ** (0.023) | | −0.042 * (0.013) | | −0.028 * (0.009) | | 0.012* (0.023) |
| Pol | | −0.027 (0.022) | | 0.037 *** (0.009) | | 0.017 *** (0.025) | | 0.036 * (0.025) |
| Cor | | −0.021 (0.022) | | 0.001 * (0.011) | | −0.007 ** (0.011) | | −0.046 ** (0.021) |
| Permit | | 0.047 * (0.024) | | 0.003 * (0.011) | | 0.024 ** (0.012) | | −0.111 *** (0.020) |
| fage | 0.022 (0.036) | 0.021 (0.035) | −0.031 *** (0.016) | −0.029 * (0.017) | 0.062 *** (0.017) | 0.059 *** (0.016) | 0.106 *** (0.031) | 0.114 *** (0.031) |
| fsize | 0.2 5*** (0.073) | 0.255 *** (0.072) | 0.261 *** (0.009) | 0.261 *** (0.009) | 0.177 *** (0.032) | 0.167 *** (0.040) | 0.147 *** (0.041) | 0.182 *** (0.033) |
| loc | 0.08 ** (0.034) | 0.011 (0.034) | −0.042 ** (0.011) | −0.042 *** (0.011) | 0.028 ** (0.012) | 0.022 * (0.012) | 0.115 *** (0.019) | 0.096 *** (0.019) |
| ctfc | 0.48 *** (0.124) | 0.484 *** (0.122) | 0.352 *** (0.029) | 0.352 *** (0.029) | 0.263 *** (0.101) | 0.186 (0.123) | 0.091 (0.068) | 0.167 ** (0.066) |
| mae | −0.04 (0.037) | −0.021 (0.037) | 0.107 *** (0.018) | 0.105 *** (0.017) | 0.032 (0.024) | 0.029 (0.022) | 0.070 *** (0.027) | 0.046 (0.030) |
| innov | 0.062 (0.056) | 0.078 (0.055) | 0.166 *** (0.035) | 0.167 *** (0.032) | 0.123 * (0.063) | 0.096 (0.068) | 0.175 ** (0.086) | 0.247 *** (0.074) |
| own | 0.348 *** (0.083) | 0.227 *** (0.083) | 0.708 *** (0.049) | 0.706 *** (0.049) | 0.478 *** (0.154) | 0.407 ** (0.179) | 0.564 *** (0.137) | 0.619 *** (0.113) |
| Const | −0.430 (0.629) | −0.399 (0.616) | −2.599 *** (0.096) | −2.603 *** (0.118) | −1.919 *** (0.451) | −1.687 *** (0.564) | −2.119 *** (0.334) | −2.078 *** (0.260) |
| Obs | 4596 | 4596 | 24,415 | 24,415 | 12,967 | 12,967 | 5344 | 5344 |

Note: Standard errors in parentheses. ***, **, * denote significance at 1%, 5%, 10%, respectively. All regressions include the year and sector fixed effects. LI, LMI, UMI, and HI are four country groups regarding the national income.

The set of control variables contributes significantly to explaining the relationship between credit constraints and export probability. Foreign-invested enterprises create favorable conditions for export development, especially in *LMI* countries. International certificate holdings and innovation tend to have a substantial impact on the export probabilities in the least developed economies (group *LI*) and medium developed countries (groups *LMI* and *UMI*). Similar to Bilkey [92], firm size has a positive effect on the ability to export. Large firms tend to export more easily than small firms. Freeman [93] believed that geographical location in large-scale cities creates many development opportunities for firms. Although the competition here can be fierce, challenges always come with opportunities if firms take advantage and develop efforts [94]. Therefore, these difficulties push firms to improve products, human resources, and management. Investments in big cities can support firms to export. However, headquarters located in big cities do not seem to bring export advantages in other countries *LMI*).

In a nutshell, this study shows that credit restriction reduces the likelihood of international market entry. Furthermore, the export disadvantage is exacerbated by institutional constraints in the *LI* and *LMI* countries. Since specifications characteristics are different across country groups, the effect of institutional obstacles and credit constraints on export status is thus different across groups.

Next, Table 5 reports the results in the further analysis when each institutional obstacles are considered separately. From the comparison between groups by institutional obstacles status, the nexus between credit constraints and exports is elucidated. In general, once each institutional obstacle is considered separately, the interaction of each institutional obstacle on the relationship between credit status and exports is more pronounced. Institutional obstacles significantly moderate the relationship between credit constraints on firms' export status in *LMI*, *UMI*, and *HI* groups, except for a few cases. However, for low-income countries, this interaction does not explain the likelihood of a firm entering the international market.

**Table 5.** The effect of credit constraints on export under each obstacles separately.

| | Tax Rate | | Political Instability | | Corruption | | Permit | |
|---|---|---|---|---|---|---|---|---|
| | No (1) | Yes (2) | No (3) | Yes (4) | No (5) | Yes (6) | No (7) | Yes (8) |
| **Panel A: LI** | | | | | | | | |
| CC | −0.091 (1.849) | −0.023 * (0.274) | 0.001 (0.430) | −0.065 (0.458) | 0.007 (0.326) | −0.053 (0.565) | −0.030 (0.270) | −0.047 * (0.899) |
| Control variables | YES | YES | YES | YES | YES | YES | YES | YES |
| Const | −1.126 (2.492) | −0.132 (0.610) | −1.515 (0.791) | −1.039 (0.979) | 0.306 (0.733) | −1.047 (0.977) | 0.102 (0.511) | −1.590 (1.523) |
| Obs | 863 | 3733 | 1573 | 3023 | 1349 | 3247 | 1808 | 2788 |
| **Panel B: LMI** | | | | | | | | |
| CC | −0.134 *** (0.304) | −0.056 ** (0.350) | −0.133 ** (0.437) | −0.055 *** (0.317) | −0.170 ** (0.488) | −0.054 ** (0.280) | −0.087 ** (0.283) | −0.058 * (0.406) |
| Control variables | YES | YES | YES | YES | YES | YES | YES | YES |
| Const | −2.624 *** (0.210) | −2.516 *** (0.105) | −2.652 *** (0.291) | −2.558 *** (0.103) | −2.564 *** (0.152) | −2.560 *** (0.152) | −2.422 *** (0.181) | −2.686 *** (0.103) |
| Obs | 6463 | 17,952 | 7571 | 16,844 | 6215 | 18,200 | 10,400 | 14,015 |
| **Panel C: UMI** | | | | | | | | |
| CC | −0.073 (0.024) | −0.075 ** (0.032) | −0.097 * (0.057) | −0.061 (0.080) | −0.055 (0.016) | −0.084 ** (0.490) | 0.028 (0.026) | −0.174 *** (0.048) |
| Control variables | YES | YES | YES | YES | YES | YES | YES | YES |
| Const | −2.260 (1.002) | −2.443 *** (0.639) | 0.269 (0.783) | −2.564 *** (0.248) | −1.403 *** (0.490) | −2.500 *** (0.711) | −1.684 *** (0.524) | −2.218 *** (0.809) |
| Obs | 4333 | 8635 | 5835 | 7133 | 6282 | 6686 | 7519 | 5449 |
| **Panel D: HI** | | | | | | | | |
| CC | 0.148 (1.059) | −0.288 *** (0.576) | −0.008 (1.155) | −0.327 *** (0.641) | −0.137 (0.995) | −0.232 *** (0.889) | −0.030 (1.013) | −0.335 *** (1.098) |
| Control variables | YES | YES | YES | YES | YES | YES | YES | YES |
| Const | −2.269 *** (0.432) | −1.995 *** (0.458) | −1.791 *** (0.301) | −2.423 *** (0.451) | −1.996 *** (0.323) | −2.691 *** (0.358) | −1.812 *** (0.246) | −2.786 *** (0.550) |
| Obs | 1086 | 4332 | 2357 | 3066 | 2961 | 2463 | 2942 | 2481 |

Note: Standard errors in parentheses. ***, **, * denote significance at 1%, 5%, 10%, respectively. "NO" means a firm does not face to a institutional obstacle. "YES" means a firm faces to a institutional obstacle. All regressions include the year and sector fixed effects. The set of control variables are fage, fsize, loc, ctfc, mae, innov, and own.

Panel A shows that in countries with an unreasonable tax rate framework (Column 2) and cumbersome licensing procedures (Column 8), easy access to credit creates an export advantage for firms. Particularly, credit-constrained firms are likely to reduce their export probabilities by 2.3% when they also face a tax rate obstacle. Meanwhile, the export probability doubled (4.7%) when that firm encountered obstacles in business licensing and permits. However, political instability and corruption cannot moderate the relationship between credit restrictions and exports, as the correlation coefficients were not statistically significant. This result was predictable, as the relationship between political instability, corruption, and credit constraints was not found in Table 1 .

Panel B reflects the negative outcome of credit constraints in the *LMI* group. The difference in the impact of each institution type on the relationship between credit restrictions and exports is almost negligible. Particularly, for firms that suffer in business licenses (Column 8), credit constraints are likely to reduce their probability of entering the international market by 5.8%. Meanwhile, this figure is 5.6% (Column 2) for firms facing tax rate obstacle, 5.5% and 5.4% for the other two types of obstacles (Columns 4 and 6). These results support for Bernard's study [39] . Firms wishing to enter export markets or expand exports may have more substantial capital requirements. These firms need to borrow more to meet their capital needs. As a result, they might struggle once rejected or only receive partial credit. Even in the case of non-credit constrained firms, the correlation coefficient of credit constraints is found to be significant. However, the presence of obstacles alleviates this negative impact. Among institutional constraints, the interaction of corruption is most effective. Specifically, credit constraints have negative impacts even in healthy institution conditions (Column 5—the probability of export decreases 17%). However, under the influence of corruption, access to credit has the weakest effect on export probability (Column 6—the probability of export decreases 5.4%). This finding supports Klapper [95]. He argued that in developing countries or where corruption is high, bribery could help firms clear up obstacles. He found the adverse effects of institutional barriers on the opportunities for firms to enter foreign markets in countries with low corruption rates. However, the regulatory obstacles have not reduced the number of firms entering the market.

Panels C and D reflect the negative results in the *UMI* and *HI* groups. The responses to export decisions when firms face credit constraints and institutional obstacles are pretty similar between these two country groups (except for the results of political instability). In particular, our results do not find a relationship between credit constraints and exports in both groups for firms with or without institutional constraints (Columns 1, 5, and 7). Meanwhile, credit constraints reduce the firm's favorable to export (Columns 2, 6, and 8). However, the results recorded in the *HI* group were nearly three times more robust than in the *UMI* group. Typically (Column 2), the export probability in the HI group is likely to be reduced by 28.8% due to credit constraints, which is three times higher than that of counterparts in *UMI* group (7.5%). These findings emphasize the importance of institutional issues in business operations. The healthy institution almost eliminated the adverse effects of credit constraints. However, when the institution begins to show weakness, inconsistency, and transparency, the disadvantages caused by credit restrictions are recognized. In a weak institution, the ability to participate in international markets is reduced when firms have difficulty accessing finance.

In summary, obstacles clustering has provided some very significant results. Regardless of institutional obstacles, for firms in *LMI* group, difficulty accessing credit is a barrier to exporting. For the rest of the countries. However, the findings indicate that credit constraints only really affect exports when institutional obstacles are controlled in the model.

In the further analysis, Table 6 presents all findings in two group by firm size. Overall, the negative consequences of credit constraints on international trade are evident. Regardless of the firm size, available external finance is an advantage for firms to enter the export market.

**Table 6.** Compare the influence of credit constraint on export by firm size.

| | SMEs | | Large-Sized Firms | |
|---|---|---|---|---|
| | **Model 3 (1)** | **Model 4 (2)** | **Model 3 (3)** | **Model 4 (4)** |
| CC | −0.163 *** | −0.166 *** | −0.106 ** | −0.104 ** |
| | (0.019) | (0.019) | (0.037) | (0.037) |
| taxr | | 0.010 * | | 0.018 |
| | | (0.008) | | (0.015) |
| pol | | 0.028 *** | | 0.000 |
| | | (0.007) | | (0.015) |
| cor | | −0.019 ** | | −0.027 * |
| | | (0.008) | | (0.015) |
| permit | | −0.009 * | | 0.016 |
| | | (0.009) | | (0.017) |
| fage | 0.032 ** | 0.032 ** | 0.013 | 0.012 |
| | (0.013) | (0.013) | (0.023) | (0.023) |
| loc | 0.076 *** | 0.076 *** | 0.062 *** | 0.062 *** |
| | (0.008) | (0.008) | (0.015) | (0.015) |
| ctfc | 0.397 *** | 0.401 *** | 0.461 *** | 0.463 *** |
| | (0.022) | (0.022) | (0.034) | (0.034) |
| mae | 0.085 *** | 0.081 *** | 0.167 *** | 0.166 *** |
| | (0.013) | (0.013) | (0.025) | (0.025) |
| innov | 0.227 *** | 0.227 *** | 0.221 *** | 0.219 *** |
| | (0.021) | (0.021) | (0.038) | (0.038) |
| own | 0.738 *** | 0.738 *** | 0.697 *** | 0.695 *** |
| | (0.029) | (0.029) | (0.041) | (0.041) |
| Const | −2.620 *** | −2.631 *** | −2.966 *** | −2.966 *** |
| | (0.066) | (0.067) | (0.163) | (0.164) |
| Obs | 40,016 | 40,015 | 7386 | 7386 |

Note: Standard errors in parentheses. ***, **, * denote significance at 1%, 5%, 10%, respectively. All regressions include the year and sector-fixed effects.

Columns 1 and Column 3 report the findings from Model 3. The variable of major interest is credit constraints, which negatively affect the ability to enter the export market of *SMEs* and large-sized firms, by 16.3% and 10.6%, respectively. These findings are not inconsistent with those of Pietrovito's study [96]. They found that credit constraints reduced the export advantage of *SMEs*. An *SME* with easy access to finance is 2.5% more likely to be an exporter than an *SME* with limited credit. Pietrovito's study covered 65 emerging and developing countries from 2003 to 2014. In comparison, our research uses a broader dataset and covers 131 countries over the recent period. This reason might explain why our coefficient is more considerable.

Columns 2 and 4 report the results from regression Model 4. Under the participation of four explanatory variables representing institutional obstacles, the correlation coefficient obtained in both groups is not much different. However, it is worth noting that the weakness of institutions makes the effect of credit constraints on exports stronger (from 16.3% to 16.6%). In addition, *SMEs* are negatively affected by inconsistencies, cumbersome mechanisms, corruption, and political instability. Meanwhile, we only find the impact of corruption on the export status of large firms at the 10% significance level. When large enterprises face institutional constraints, credit-constrained firms reduce the probability of exporting by 0.1% compared to their counterparts.

In addition, a firm's characteristics also contribute to the impact on the firm's ability to export, regardless of firm size. Firm age is particularly significant for *SMEs*. As previously argued by Beck [53] and Shinkle [97], it is difficult to export for a small and young firm

located in small cities. Furthermore, innovation and experience also play essential roles in explaining export advantages.

In summary, Analysis 2 was carried out to find the relationship between credit constraints and export and the role of institution obstacles in this linkage. The findings show that firms' access to capital negatively affects exports in all regions. The results in the group of rich countries are most pronounced. In addition, institutions' health exacerbates the impact of credit restrictions on exports. Firms that operate in weak institutions are also less likely to participate in exports. Moreover, *SMEs* suffer the consequences of credit constraints more clearly than large-sized firms. In addition, this is also a vulnerable group of firms in a weak institution.

## 5. Conclusions

The credit issue is a concern for both large-sized firms and *SMEs*, especially in places where capital markets are not yet completely developed, such as developing countries or emerging markets. This study analysis the effect of institutional obstacles on credit constraints and the relationships between credit constraints and export decisions, controlling the interaction of institutional obstacles. The regression results confirm the importance of institutions to a firm's credit status. It is difficult for firms to meet the standards of formal credit organizations under strict Goverment's control over tax rate, political instability, corruption, and business licenses. As a result, credit constraints become more serious. A weak institutionsincreases the credit constraints and affects the export situation of firms. Credit-constraintsed firms reduced ability to participate in international markets.

Our results suggest a number of policy-related issues. To promote exports as well as create facilitate for firms, policymakers might consider taking specific measures to support firms in overcoming financial difficulties [98,99]. For example, lending incentives can be made possible by loosening lending regulations and conditions. Diversifying funding sources (banks, financial firms, and other funding sources) is also a viable solution. In addition, to ensure the supply of capital, the government might develop specific support programs for credit orgnizations and banks. Besides, for low-income countries with a high corruption index, policies should prioritize cleaning up the bureaucracy. Restructure the management apparatus and apply transparent management measures to reduce the phenomenon of corruption in various forms. Furthermore, during the political transition, such as changing ruling parties and preparing for elections, policies to support business need to be implemented. These measures can help firms reduce the pressure of limited access to external capital [68,69]

Besides contributing to policymakers' visions, this study helps a firm recognize the importance of external capital and its characteristics that should be improved. In particular, corporate restructuring is becoming a top concern under the pressure of economic integration. Restructuring can be done on an annual, cyclical basis. This solution might help firms to conduct a continuous internal review. Restructuring is a crucial factor in helping a firm improve operational efficiency and minimize costs to deal with consequences. Moreover, most loans use the land as reliable collateral. Because land is an item belonging to the State, it is easier for credit institutions to control information, thereby reducing loan risk [100]. Besides, the land also shows high-value and available assets that a firm own. Therefore, using land as collateral is a crucial determinant of firms' access to credit [101]. Therefore, firms, especially *SMEs*, should focus more on investing and holding tangible assets (land) to diversify collateral. From there, the ability to match loan conditions can be improved.

**Author Contributions:** Conceptualization, T.H.P., R.S. and H.T.H.N.; methodology, T.H.P.; software, T.H.P.; formal analysis, T.H.P.; data curation, T.H.P.; writing—review and editing, T.H.P., R.S., and H.T.H.N.; supervision, R.S. and H.T.H.N. All authors have read and agreed to the published version of the manuscript.

**Funding:** We acknowledge support by the Deutsche Forschungsgemeinschaft (DFG—German Research Foundation) and the Open Access Publishing Fund of Technical University of Darmstadt - Project number 413727769.

**Institutional Review Board Statement:** Not applicable.

**Informed Consent Statement:** Not applicable.

**Data Availability Statement:** All data generated or analyzed to support the findings of the present study are included this article. The raw data can be obtained from the authors, upon reasonable request.

**Conflicts of Interest:** The authors declare no conflict of interest.

## Appendix A

**Table A1.** The list of countries.

| Low-Income Countries | Low-Medium-Income Countries | Upper-Medium-Income Countries | High-Income Countries |
|---|---|---|---|
| Afghanistan | Angola | Albania | Barbados |
| Burkina Faso | Bangladesh | Argentina | Belgium |
| Burundi | Benin | Armenia | Chile |
| Central African Republic | Bhutan | Azerbaijan | Croatia |
| Chad | Bolivia | Belarus | Cyprus |
| DRC | Cambodia | Bosnia and Herzegovina | Czech Republic |
| Eritrea | Cameroon | Botswana | Estonia |
| Ethiopia | Cape Verde | Brazil | Greece |
| Gambia | Congo | Bulgaria | Hungary |
| Guinea | Côte d'Ivoire | Bulgaria | Israel |
| Liberia | Djibouti | China | Italy |
| Madagascar | Egypt | Colombia | Latvia |
| Malawi | El Salvador | Costa Rica | Lithuania |
| Mali | Ghana | Dominica | Luxembourg |
| Mozambique | Honduras | Dominican Republic | Malta |
| Niger | India | Ecuador | Mauritius |
| Rwanda | Kenya | Gabon | Panama |
| Sierra Leone | Kyrgyzstan | Georgia | Poland |
| South Sudan | Lao PDR | Guatemala | Portugal |
| Sudan | Lesotho | Guyana | Romania |
| Tajikistan | Mauritania | Indonesia | Slovakia |
| Togo | Moldova | Iraq | Slovenia |
| Uganda | Mongolia | Jamaica | Sweden |
| Yemen | Morocco | Jordan | Trinidad and Tobago |
| | Myanmar | Kazakhstan | Uruguay |
| | Nepal | Lebanon | |
| | Nicaragua | Malaysia | |
| | Nigeria | Mexico | |
| | Pakistan | Montenegro | |
| | Papua New Guinea | Namibia | |
| | Philippines | North Macedonia | |
| | Senegal | Paraguay | |
| | Sri Lanka | Peru | |
| | Tanzania | Russia | |
| | Timor-Leste | Serbia | |
| | Tunisia | South Africa | |
| | Ukraine | Suriname | |
| | Uzbekistan | Thailand | |
| | Vanuatu | Turkey | |
| | Vietnam | Venezuela | |
| | Zambia | | |
| | Zimbabwe | | |

Note: Country classification by income based on World bank's publication 2020.

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
