# Peer review of "Export Decision and Credit Constraints under Institution Obstacles"

_sustainability, doi:10.3390/su14095638_

Round 1
Reviewer 1 Report
The topic chosen by the authors is interesting and analyzed at both theoretical and practical levels. The article structure meets the scientific criteria for publication in the journal. Despite this, there are some aspects that the authors should pay attention to:
- The Abstract could be improved. The research goal is not clearly defined and not all methods used are mentioned.
- Keywords could be more concrete and specific.
- The introduction is too long. It would be better to divide it into 2 chapters: Introduction and Literature Review. Please highlight the novelty and originality of your research.
- In page 4, you have written: “Since the dependent variables are binary, I apply the Probit method to the models”. What do you mean by “I”? How many authors there are of this article?
- Some of the information in the “Data and Measurement Credit constraints” could be moved to the literature review part.
- The data chapter should be concentrated on what data you have used and what methods you have applied for your research. Could you describe in detail what are the main steps of the research? On page 11 you have written: “In the second step, I <…>”.
- In the scientific discussion part, I would advise you to write, how your research results differ or correlate with the results of previous studies.
- There could be more conclusions from the empirical part of your research.
RECOMMENDATION: REVISIONS REQUIRED
Author Response
Dear Sir/Madam,
First of all, I would like to thank you very much for giving us the opportunity to respond to your comments. Your insightful reviews and suggestions help us look at issues more holistically. We have accepted your comments and revised our manuscript. We hope that our efforts can solve the problems you have raised. Given the limitations of time and data, some problems cannot be fully resolved. We appreciate your comments, and we will certainly complete them in future work.
Here, I list the issues you raised, along with our detailed responses point by point. To be more specific, you can see the manuscript file.
Point 1. The Abstract could be improved. The research goal is not clearly defined and not all methods used are mentioned.
Response: Following your comment. I re-write the abstract as below:
"Abstract: The growing demand for goods and technology increases capital requirements, especially in exporting enterprises. However, many enterprises have difficulty accessing external capital due to institutional obstacles. This study analyzes two main issues: the influence of institutional obstacles on credit constraints and the relationship between credit constraints and export decisions, adopting firm-level data from 131 countries.
The study's remarkable contribution was to cluster the data into four country groups based on their national income. The typical specification of each group can lead to more precise results, thereby highlighting the role of institutions. More advanced, this study complements the literature's gap in the relationship between credit restrictions and exports by controlling for institutions as interactive variables in the model. This work upgrades assessments to be more accurate, thereby providing more valuable information to policymakers. Besides, credit constraints are measured by both quantitative and qualitative methods. The essential role of firm size is emphasized in further analysis. This study approaches the Probit method. Furthermore, an instrumental variable is used to solve the endogeneity problem.
The results found that a weak institution prevents access to finance, especially in middle-income countries. In addition, firms' access to capital negatively affects exports in all regions. The finding in the group of rich countries is most pronounced."
Point 2: Keywords could be more concrete and specific.
Response: I re-write the abstract as below:
"Keywords: Credit constraints, Export, Institutions, Tax rate, Political instability, Corruption, Business licensing and permit, World Bank data 2020, IV-Probit regression”
Point 3. The introduction is too long. It would be better to divide it into 2 chapters: Introduction and Literature Review.
Response: Following your suggestion, I have separated the introduction into two parts: Introduction and Literature Review.
Point 4. Please highlight the novelty and originality of your research.
Response: I highlight the novelty and originality of my study in line 167 -198
" This study makes several outstanding contributions.
First, this study fills a gap in the literature on firm-level financial constraints. Previous studies analyzed financial constraints in many aspects, such as financial leverage, liquidity ratios, and overdrafts, but few studies focused on credit constraints. Significantly, the intertwined constraints between institutions, credit obstacles, and exports have not been considered. Unlike earlier studies, this study sheds light on the effect of institutional obstacles on credit constraints and scrutinizes its impact on the relationship between credit constraints and participating in the international market. Analyzing the interaction of institutions is necessary to perceive the social relationships and firm performance correctly.
Second, there has been no research in the literature on the impact of institutional obstacles (including tax rate, political instability, corruption, and licensing) on credit restriction. Especially, the study utilizes the World Bank firm-level data cross-countries. Furthermore, the paper provides a visualization of the marginal impact of institutional obstacles on credit constraints in two groups, including exporters and non-exporters. Therefore, our study might fill in this missing information. The importance of institutional constraints on firm performance is exposed. Based on that, the government can consider appropriate institutional reform strategies to encourage business development.
Third, the remarkable contribution of the study is to cluster the data into four groups of countries based on their national income. Institutions and national growth appear to be linked \mbox{\cite{Easterly1997}}. Firms in the same group with typical institutional specifications might have similar responses to credit constraints and exports. Thus, grouping can bring more accurate results, thereby highlighting the role of institutions. As a result, the findings might provide ideas for managers and policymakers in specific areas. Through it, appropriate strategies for the unique characteristics of each economy can be designed.
Four, this study briefly reviews the credit constraint measurements. A quantitative method through several criteria in the enterprise's financial statements is applied in the main analysis. This method is more thorough than others because it combines information about a firm's credit line, declined debt, and credit needs. At the same time, qualitative measurement methods are also approached in the robustness test section.
Finally, this study conducts further sensitive analyses into the relationship between credit constraints and exports. In particular, each type of institutional constraint is observed separately. Besides, the paper carried out tests on two groups of enterprises classified by size.”
Point 5. In page 4, you have written: "Since the dependent variables are binary, I apply the Probit method to the models". What do you mean by "I"? How many authors there are of this article?.
Response: Thank you. This is a mistake. On behalf of the authors, I corrected it.
Point 6. Some of the information in the "Data and Measurement Credit constraints" could be moved to the literature review part.
Response: I moved a paragraph about the previous studies on measurement credit constraints from "Data and Measurement Credit constraints" into the literature review part. Now, this information placed in line 224 - 294
Point 7. The data chapter should be concentrated on what data you have used and what methods you have applied for your research. Could you describe in detail what are the main steps of the research?
Response: I have rearranged the contents of the Data section. In particular, I moved the Data source part to Section 3: Empirical specification. Then, I moved paragraphs related to previous studies to the Literature Review section (Section 2) (from line 224 - 294). Besides, I moved the rest regarding the measurement of Credit constraints, which will be used in this study, to the section Measuring credit constraints in Section 3: Empirical specification (line 464).
In brief, the main steps of our research are:
- Analysis 1: The effect of institutional obstacles on credit constraints.
- First, we regress Model 1 and 2 on four country groups by approaching Probit regression. The credit constraints variable is measured by a quantitative method (denoted as CC).
Ins_ijs={tar,permit,pol,cor}$ is the set of institutional obstacles.
- Then, we report the predicted marginal effects of four institutional obstacles in four country groups on credit constraints at the 95\% confidence interval. We compare two groups by export status
- In the robustness test, instead of quantitative credit constraints (CC) we approach qualitative credit constraints (denoted as FIN)
- Analysis 2: The linkage between credit constraints and export decision.
- First, we regress Model 3 on four country groups to examine the relationship between credit constraints (denoted as CC) and export decisions. We use the IV-Probit regression to solve the endogenous problem of credit constraint. We use the information about corporate audits as an instrumental variable (named audit).
- Then, we regress Model 4 on four country groups. We put the institutional obstacles variable (marked as Ins_ijs) on the Model 4 to examine the relationship between credit constraints (denoted as CC) and export decisions under the effect of institutional obstacles. Ins_ijs={tar,permit,pol,cor}$ is the set of institutional obstacles. IV-Probit regression with audits as an instrumental variable.
- Next step, each type of institutional obstacle is examined individually. We separate each institutional obstacle into two groups. One group consists of firms that face an institutional obstacle (at minor, moderate, major, and very severe levels). Another group includes no institutional obstacle firms. Then we examine the relationship between credit constraint and export in two above groups for each institutional obstacle. IV-Probit regression with audits as an instrumental variable.
- Finally, we separate firms into two groups by size, including SMEs and large-sized firms. Then we regress Model 3 and 4. IV-Probit regression with audits as an instrumental variable.
Point 8. On page 11 you have written: "In the second step, I <…>".
Response: I corrected it
Point 9. In the scientific discussion part, I would advise you to write, how your research results differ or correlate with the results of previous studies
There could be more conclusions from the empirical part of your research
Response: We restructure the discussion and conclusion. Specifically, in section 4, we report the study results and discuss them. Part 5 is the conclusion.
In Section 4, we added more discussion related to the study results and connected them with previous studies. Since revisions are often made, please see the pdf for details, section 4.
Once again, thank you for your time and your support.
Best regards.

Reviewer 2 Report
This is an interesting proposal. However, the authors must address several issues prior to publication:
- The use of the English language must be improved; there are serious problems with the proposed article from this point of view.
- Terminology: the authors use the term “institution” with different meanings throughout the article. The terms “institutional environment”, “limitations”, “factors”, “barriers”, “obstacles”, etc. are apparently used interchangeably, which can confuse the reader. In addition, the phrase “credit institution” is also used. In addition, a definition of institution is missing. Authors must provide a definition of institution and use only one term throughout the article. The reference to North (line 509 in the manuscript) should appear earlier in the article. For clarity, "credit institution" should be replaced by a different term, such as "credit provider".
- The authors should clarify that Section 2, “Data and Measurement Credit Constraints” is in fact a review of the existing literature on this topic. Here, the term “chapter” (line 174) should be replaced by “section”.
- Section 3, “Empirical Specification” should be divided into Methodology and Findings. Here, it would be convenient to clarify the meaning of the phrase “Companies in weak institutions” (line 647) and others similar to it.
- The section devoted to discussion and conclusions should be drastically expanded and improved. As it stands, this section comprises very few references, as well as unreferenced and hazardous statements (such as "credit institutions also need to receive support from the government " – line 674, or "political change has had a rather dramatic effect on rich nations" – line 679). Authors should stick to the conclusions that follow directly from their findings. The political implications of the findings should also be emphasized. For the overall balance of the article, the Methodology and Findings sections should be reduced in length.
If the authors can address the above problems, the result should be an interesting and useful article.
Author Response
Dear Sir/Madam,
First of all, I would like to thank you very much for giving us the opportunity to respond to your comments. Your insightful reviews and suggestions help us look at issues more holistically. We have accepted your comments and revised our manuscript. We hope that our efforts can solve the problems you have raised. Given the limitations of time and data, some problems cannot be fully resolved. We appreciate your comments, and we will certainly complete them in future work.
Here, I list the issues you raised, along with our detailed responses point by point. To be more specific, you can see the manuscript file.
Point 1: Terminology: the authors use the term “institution” with different meanings throughout the article. The terms “institutional environment”, “limitations”, “factors”, “barriers”, “obstacles”, etc. are apparently used interchangeably, which can confuse the reader. In addition, the phrase “credit institution” is also used. In addition, a definition of institution is missing.
Authors must provide a definition of institution and use only one term throughout the article.
The reference to North (line 509 in the manuscript) should appear earlier in the article. For clarity, "credit institution" should be replaced by a different term, such as "credit provider".
Response:
- Following your comments, we changed confusing terminology. We use “credit constraints”, “institutional obstacles”, and “institutions” throughout the article.
- We added the definition of institution and a weakness institution on Line 338-354:
“Notably, many scientists emphasized that political and institutional factors bind many economic and social phenomena. In other words, institutional specifications are likely to bias relationships in fields (Coase1937). Therefore, controlling for institutional characteristics is the key to analyzing relationships realistically and deeply. The terminology "institutions" was first mentioned by Thorstein Veblen in 1898, after which it has been widely applied to explain many behavioral phenomena and the decision-making processes of managers. Accordingly, the concept of "institution" includes regulations, principles, and laws used to regulate the activities of individuals and organizations. North1989 argued that the institutional framework consists of formal and informal institutions. Authorities implement official rules, including constitutions, statutes, charters, and legal documents.
Meanwhile, informal rules can expand and govern subjects' behavior through social norms (traditions, beliefs, customs, taboos, etc.). Empirical researchers proposed several indicators to assess the institutions' quality, including political instability, corruption, political regime characteristics, social management (such as constitution, laws, procedures, and regulations), etc. A weak institution is characterized by political instability, severe corruption, suboptimal legality, and ineffective enforcement of regulations. On the contrary, it can be called a healthy institution.”
- We replaced the term “credit institutions” to “credit organization” and “lender”.
Point 2: The authors should clarify that Section 2, “Data and Measurement Credit Constraints” is in fact a review of the existing literature on this topic.
Here, the term “chapter” (line 174) should be replaced by “section”.
Response:
- We replaced the term “chapter” (line 174) to “section”.
- I have rearranged the contents of the Data section. In particular, I moved the Data source part to Section 3: Empirical specification. Then, I moved paragraphs related to previous studies to the Literature Review section (Section 2) (from line 224 - 294). Besides, I moved the rest regarding the measurement of Credit constraints, which will be used in this study, to the section Measuring credit constraints in Section 3: Empirical specification (line 464).
Point 3: Section 3, “Empirical Specification” should be divided into Methodology and Findings.
Here, it would be convenient to clarify the meaning of the phrase “Companies in weak institutions” (line 647) and others similar to it
Response: We separated the findings into a new section (section 4: Estimation Results and Discussion).
Now, Section 3 “Empirical Specification” presents the data source, the method to measure credit constraints that use in this paper, and strategies.
Point 4: The section devoted to discussion and conclusions should be drastically expanded and improved.
As it stands, this section comprises very few references, as well as unreferenced and hazardous statements (such as "credit institutions also need to receive support from the government " – line 674, or "political change has had a rather dramatic effect on rich nations" – line 679).
Authors should stick to the conclusions that follow directly from their findings. The political implications of the findings should also be emphasized.
For the overall balance of the article, the Methodology and Findings sections should be reduced in length.
If the authors can address the above problems, the result should be an interesting and useful article.
Response: We restructure the discussion and conclusion. Specifically, in section 4, we report the study results and discuss them. Part 5 is the conclusion. We would like to give some policy indications in the conclusion part.
In Section 4, we added more discussion related to the study results and connected them with previous studies. Since revisions are often made, please see the pdf for details, sections 4 and 5.
Once again, thank you for your time and your support.
Best regards.

Reviewer 3 Report
The title is consistent with the content presented and the abstract presents the objective, method and conclusions. However, it is necessary to better structure the abstract.
The abstract must be structured: Purpose: (text...); Results and contributions: (text...); Methodology: (text...); Gap: (text...); Relevance: (text...); Impact: (text...).
The reading of the text is fluid and the coordination of ideas is well organized.
Significant of the paper:
This paper contains sufficiently new and significant knowledge. The purpose of the investigation is relevant.
Literature:
The paper is well related to previous literature. The author demonstrates extensive knowledge of previous studies. The author should do an additional task of searching for recent work and incorporating innovative arguments in his research.
Methodology:
The research method is appropriate and suggestions for future research are presented (although they should be deepened/developed).
Results:
The research results are presented clearly and are similar to those obtained in the previous literature. However, they make an essentially descriptive analysis of the results obtained.
Limitations, contributions and suggestions for future research could be further developed.
Implications for research:
The implications is very relevant both for companies in general and for society.
Author Response
Dear Sir/Madam,
First of all, I would like to thank you very much for giving us the opportunity to respond to your comments. Your insightful reviews and suggestions help us look at issues more holistically. We have accepted your comments and revised our manuscript. We hope that our efforts can solve the problems you have raised. Given the limitations of time and data, some problems cannot be fully resolved. We appreciate your comments, and we will certainly complete them in future work.
Here, I list the issues you raised, along with our detailed responses point by point. To be more specific, you can see the manuscript file.
Point 1: The title is consistent with the content presented and the abstract presents the objective, method and conclusions. However, it is necessary to better structure the abstract.
The abstract must be structured: Purpose: (text...); Results and contributions: (text...); Methodology: (text...); Gap: (text...); Relevance: (text...); Impact: (text...).
Response: I sincerely appreciate your valuable comments. Following your guide, I re-wrote the abstract as below:
" The growing demand for goods and technology increases capital requirements, especially in exporting enterprises. However, many enterprises have difficulty accessing external capital due to institutional obstacles. This study analyzes two main issues: the influence of institutional obstacles on credit constraints and the relationship between credit constraints and export decisions, adopting firm-level data from 131 countries.
The study's remarkable contribution was to cluster the data into four country groups based on their national income. The typical specification of each group can lead to more precise results, thereby highlighting the role of institutions. More advanced, this study complements the literature's gap in the relationship between credit restrictions and exports by controlling for institutions as interactive variables in the model. This work upgrades assessments to be more accurate, thereby providing more valuable information to policymakers. Besides, credit constraints are measured by both quantitative and qualitative methods. The essential role of firm size is emphasized in further analysis. This study approaches the Probit method. Furthermore, an instrumental variable is used to solve the endogeneity problem.
The results found that a weak institution prevents access to finance, especially in middle-income countries. In addition, firms' access to capital negatively affects exports in all regions. The finding in the group of rich countries is most pronounced."
Once again, thank you for your time and your support.
Best regards.

Reviewer 4 Report
The text is interesting, however, the limitations of the research should be described very broadly (especially since the correlations found are at a low level).
It is worth considering additional ordering of the text structure, because some threads repeat excessively. For example, the discussion is spread over most of the text instead of being concentrated at the end of the article.
Minor linguistic errors should also be corrected (maybe it would also be better if the text was not written from the first person perspective) and factual errors such as:
-Page 6. Please correct ("0" and "0"?):
"I create a dummy variable named as Demand that gets a value 0 if a firm doesn’t apply for a loan because it does not have demand. Otherwise, it gets a value of 0 if a firm doesn’t apply for a loan because of another reason."
-Page 13. Please corret ("non" and "non"?):
"...for both non-exporting and non-exporting groups (Fig 1-1 and 1-2)..."
Author Response
Dear Sir/Madam,
First of all, I would like to thank you very much for giving us the opportunity to respond to your comments. Your insightful reviews and suggestions help us look at issues more holistically. We have accepted your comments and revised our manuscript. We hope that our efforts can solve the problems you have raised. Given the limitations of time and data, some problems cannot be fully resolved. We appreciate your comments, and we will certainly complete them in future work.
Here, I list the issues you raised, along with our detailed responses point by point. To be more specific, you can see the manuscript file.
Point 1: The text is interesting, however, the limitations of the research should be described very broadly (especially since the correlations found are at a low level).
Response: We have rewritten the results and discussion sections. Thereby, we reported relevant results, and provided specific discussions. For example, suggesting causation, linking to the results of previous studies. Since the amount of revisions in this section is quite large, please see the manuscript file.
We made an effort to account for unexpected results where the correlation coefficient was not statistically significant. Even so, due to time limitations, there are still some issues we haven't explained thoroughly. We will address those points thoroughly in future studies.
Point 2: It is worth considering additional ordering of the text structure, because some threads repeat excessively.
For example, the discussion is spread over most of the text instead of being concentrated at the end of the article.
Response: Thank you so much for your comment. We have restructured the article according to the comments
Point 3: Minor linguistic errors should also be corrected (maybe it would also be better if the text was not written from the first person perspective) and factual errors such as:
-Page 6. Please correct ("0" and "0"?):
"I create a dummy variable named as Demand that gets a value 0 if a firm doesn’t apply for a loan because it does not have demand. Otherwise, it gets a value of 0 if a firm doesn’t apply for a loan because of another reason."
-Page 13. Please corret ("non" and "non"?):
"...for both non-exporting and non-exporting groups (Fig 1-1 and 1-2)..."
Response: We corrected them.
Once again, thank you for your time and your support.
Best regards
